# JLH Mark2 - An Improved Opto-Mechanical Approach to Open-Path *in situ* Water Vapor Measurement in the Upper Troposphere / Lower Stratosphere

Robert L. Herman[1], Robert F. Troy[2,3], Kim M. Aaron[1], Isabelle Sanders[1], Kevin Schwarm[1], Joshua Eric Klobas[4], Aaron Swanson[4], Andrew Carpenter[4], Scott Ozog[4], Keith Chin[3], Lance E. Christensen[1], Dejian Fu[1], Robert F. Jarnot[1], Robert A. Stachnik[3], Ramabhadran Vasudev[3]

[1]Jet Propulsion Laboratory, California Institute of Technology, 4800 Oak Grove Drive, Pasadena, California, 91109, USA.
[2]Robert Troy Engineering, West Hills, California, 91307, USA.
[3]Formerly at Jet Propulsion Laboratory, Pasadena, California, 91109, USA.
[4]Northrop Grumman Systems Corporation, Redondo Beach, California, 90278, USA.

*Correspondence to*: Robert L. Herman (Robert.L.Herman@jpl.nasa.gov)

**Abstract.** To improve the accuracy and precision of in situ water vapor measurements from aircraft, a new opto-mechanical design was implemented on the JPL Laser Hygrometer Mark2. The first JPL Laser Hygrometer (JLH Mark1), originally developed in mid-1990s, provided airborne in situ water vapor measurements for fifteen years from several platforms, including the NASA ER-2 and WB-57 aircraft. Due to heavy use over the years and aging of the instrument parts, many of the components in JLH Mark1 have been modified and replaced. This instrument paper reports on the redesigned opto-mechanical structure of the instrument, new data retrieval algorithms, and updated data analysis procedures, along with recent laboratory and field performance and a comparison with another water vapor instrument. Key changes in the redesigned instrument have significantly improved the performance, as demonstrated during the NASA Studies of Emissions and Atmospheric Composition, Clouds and Climate Coupling by Regional Surveys (SEAC[4]RS) field mission and eight years of subsequent science flights on the Northrop Grumman Systems Corporation Flying Test Bed (FTB).

## 1 Introduction

Atmospheric measurements of water vapor in the upper troposphere and lower stratosphere (UTLS) are important for better understanding Earth's radiative balance because UTLS water has a significant impact on both radiative forcing and stratospheric ozone despite its low concentrations. Through the radiative impact of water vapor (Solomon et al., 2010) and cirrus clouds, climate is strongly affected by atmospheric water. Clouds continue to be one of the biggest sources of uncertainty in climate prediction (Boucher et al., 2013, Davis et al., 2007). An additional climate contribution from clouds is due to aircraft. In the upper troposphere, significant fractions of the area are ice supersaturated regions (e.g., Troy, 2007; Gettelman et al., 2006; Heymsfield et al., 1998). In such an environment, aircraft engine combustion elevates the relative humidity high enough for homogeneous ice nucleation in persistent contrails and aircraft induced cirrus clouds (e.g., Testa et al., 2024, and references therein). Burkhardt and Kärcher (2011) found that aircraft induced cirrus are the largest net (warming) effective radiative forcing term from aviation, larger even than the radiative forcing term from $CO_2$ emissions from aircraft (Lee et al., 2021). There is a renewed interest in minimizing contrails in commercial aviation, which has led to the aircraft campaign coordinated by the Northrop Grumman Systems Corporation (NGSC) and described herein.

The JPL Laser Hygrometer (JLH) for *in situ* UTLS water vapor measurements has been developed and tested over many years. The initial version JLH Mark1(a) instrument provided airborne measurements from the NASA ER-2 aircraft (May, 1998). Several years later, a similar JLH Mark1(b) was developed by R. D. May at JPL for the NASA WB-57F aircraft. Both instruments are open-path near-infrared tunable diode laser absorption spectrometers designed to accurately measure atmospheric water vapor concentration from aircraft. The two JLH Mark1 instruments were among the first laser hygrometers developed for airborne measurements and had a fifteen year history of aircraft flights in NASA UTLS field missions (see Table 1) for scientific studies of atmospheric water vapor, clouds and UTLS transport (e.g., Hintsa et al., 1999; Hurst et al., 1999; Kley et al., 2000; Gates et al., 2002; Danilin et al., 2003; Herman et al, 2003; Lee et al., 2004; Hallar et al., 2003; Ray et al., 2004; Ridley et al., 2004; Gao et al., 2004; Jensen et al., 2005; Garrett et al., 2005; Gao et al., 2006; Richard et al., 2006; Gensch et al., 2008; Froyd et al., 2010; and Herman et al., 2017).

Due to heavy use over the years and aging of the instrument parts, many of the components in JLH Mark1 have been modified and replaced. For example, extensive exposure to rocket plumes deteriorated the mirror coatings so that it became necessary to replace the mirrors in June 2007 prior to the NASA TC[4] mission (Table 1). In addition, several epoxy bonds between the original mirrors and their mounts separated at low temperature. Alternate optomechanical mounting designs were attempted during the NASA GloPac (Hintsa et al., 2010) and MACPEX (Rollins et al., 2014) field missions, but did not provide adequate thermal stability at stratospheric temperatures below 240 K.

As reported in this paper, an entirely new optomechanical approach has been implemented for JLH Mark2 (see Sections 2 and 4). We also redesigned and ruggedized the instrument, developed new data retrieval algorithms and updated the calibration and data analysis procedures. These improvements in the new JLH Mark2 version of the instrument, along with recent field and laboratory performance are described in this paper. The rest of this paper is organized as follows. Section 2 provides an outline of the instrument. Instrumental issues in JLH Mark1 are described in Section 3 and improvements incorporated in JLH Mark2 are described in Section 4. Spectroscopy and data analysis are in Section 5. New performance specifications are in Section 6, followed by a summary in Section 7.

## 2. Instrument Overview

The JLH instruments are based on the measurement of water absorption in the 1370 nm wavelength region with an open-path multi-pass cell, as described previously (May, 1998). The measurement method is 2f wavelength modulation spectroscopy (May and Webster, 1993; May, 1998). The light source is a commercial, narrow-linewidth, distributed feedback (DFB) tunable diode laser (TDL). The specific water line targeted in JLH Mark2 atmospheric measurements has a line center wavelength of 1369.97 nm ($7299.4311$ cm$^{-1}$), as shown in Table 2. The total absorption path length is 10.479 m obtained by multi-passing the laser light between the mirrors of the open-path sample cell (Section 2.1). Examples of similar tunable laser hygrometers include the NASA LaRC Diode Laser Hygrometer or DLH (Diskin et al., 2002; Podolske et al., 2003), the Harvard Herriott Hygrometer or HHH (Sargent et al., 2013), the VCSEL hygrometer (Zondlo et al., 2010), and the SEALDH-II hygrometer (Buchholz and Ebert, 2018). A core set of hygrometers including JLH Mark1 were intercompared in the AIDA chamber at the Karhsruhe Institute of Technology in the AquaVIT-1 campaign (Fahey et al., 2014). A review of modern hygrometers is given in Bange et al. (2013).

The first flights of the newly improved JLH Mark2 were on the NASA ER-2 high-altitude aircraft in 2013 during the field mission for the NASA Studies of Emissions and Atmospheric Composition, Clouds and Climate Coupling by Regional Surveys (SEAC[4]RS) (Herman et al., 2017). For these flights, JLH Mark2 was mounted in a camera port in the lower Q-bay of the NASA ER-2 fuselage (Figure 1a and Section 4). The ethernet-based NASA Airborne Science Data and Telemetry System (NASDAT) on the NASA ER-2 aircraft (Sorenson et al., 2012) was used to acquire measurements and data that included time, GPS coordinates and static

temperature/pressure from the Meteorological Measurement System (MMS; Scott et al., 1990).

Subsequently, JLH Mark2 flew on the Flying Test Bed (FTB) aircraft of the Northrop Grumman Systems Corporation for contrail studies and flights of opportunity (2015-2023). For these flights, JLH Mark2 was mounted on top of the aircraft fuselage where it samples the atmospheric freestream (Figure 1b). Northrop Grumman operates an Experimental Gulfstream II Test Bed aircraft

(N82CR) used to assess mission system payloads prior to customer delivery. This aircraft is equipped with onboard equipment racks, electrical power dedicated to instrumentation, and external hardpoints for payloads. For this effort, an additional Total Air Temperature (TAT) probe, Goodrich P/N: 102DB1CK, was integrated. The selected probe is identical to the production Gulfstream II TAT probe and was mounted in a mirrored location compared to the production, i.e. same fuselage station and vertical waterline coordinate, but on the port side of the fuselage rather than starboard. The selected TAT probe is a dual mandrel design, three inches

in length to measure outside the boundary layer, and resistively heated to prevent ice buildup. The probe was routed to an instrumentation system for data collection. Static pressure was also sampled outside the boundary layer. The airframe static pressure ports connect to Gulfstream Air Data Computers that drive an ARINC-429 (Aeronautical Radio, Incorporated) digital information transfer system (Martinec et al., 2015) which is then displayed to the pilots. The ARINC signal was split and connected to mission-system computers so that static pressure was also collected for the payloads and instrumentation. An inertial navigation

system (VectorNav P/N: VN-300) supplied position and time coordinates.

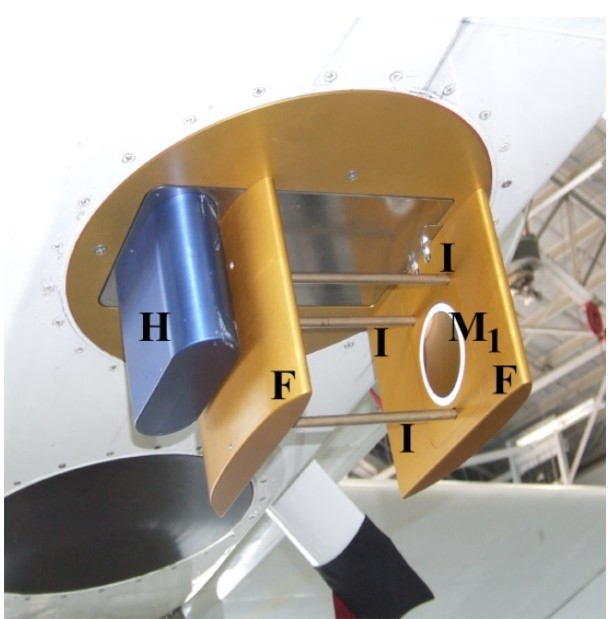 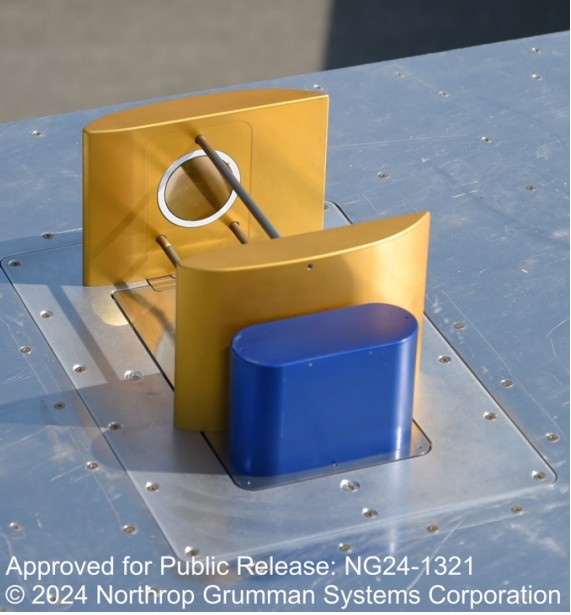

**Figure 1.** (Left) JLH Mark 2 mounted on a lower camera port underneath the NASA ER-2 fuselage Q-bay. The abbreviations used are H: evacuated housing for the tunable laser and detector; M1: the far plano-concave mirrors of the Herriott cell (the other mirror is hidden from view in this photo); I: three Invar rods used for maintaining the two mirrors at fixed separation (Section 3.1); and F: aerodynamic faring for deflecting the airflow around the mirrors of the Herriott cell, as described later in Section 4.2. (Right) JLH Mark 2 mounted above fuselage of NGSC Gulfstream-II aircraft with the same JLH structure.

The JLH instrument payload ingested these temperature, pressure, and spatiotemporal data, providing water vapor mixing ratio at 1 Hz. For a limited number of flights, a commercial hygrometer (FLYHT P/N: WVSS-II) collected data simultaneously. Throughout the JLH campaign comprising 109 total flights, JLH data were collected at pressure-altitudes spanning 10,000 ft to 45,000 ft (3000 m to 13,700 m), primarily in the daytime over the Isabella, Owens, and Bishop Military Operations Areas (MOAs) within the R-2508 complex in the Mojave Desert as well as a limited number of missions off the coast of Florida.

## 2.1 Optics

The optical open-path multipass absorption cell of JLH Mark2 is mounted external to the aircraft and situated beyond the expected boundary layer in order to avoid contamination or outgassing from surfaces (Figure 1a and 1b). The laser beam emerging from the TDL is collected by an aspheric lens and focused approximately at the midpoint of the multipass cell in the Herriott optical configuration (Altmann et al., 1981; Herriott et al., 1964). The multipass cell consists of two plano-concave mirrors made of Zerodur (lithium aluminosilicate glass-ceramic) substrate with their concave surfaces coated with gold to provide high reflectivity and an $Al_2O_3$ over-coating for protection. The mounting of the cavity mirrors in the Herriott cell is shown in the computer-aided design (CAD) drawing in Figure 2(a). The focused laser beam enters the cell through a small orifice in one of the mirrors $M_0$ (Figure 2(a)). The two mirrors ($M_0$ and $M_1$) are separated by 207.9 mm, have a diameter of 63.5 mm and radius of curvature of 219.66 mm. The angle at which the laser light is injected into the cell is such that the light travels fifty times between the mirrors before exiting the cell through the same orifice as the entrance beam. Upon exiting, it is focused onto a miniature InGaAs detector, located adjacent to the TDL. Both the laser and the detector are located inside an evacuated housing (H in Figure 1(a)), equipped with a transparent window. There is an additional 86-mm path through ambient air between the Herriott cell near mirror ($M_0$) and the window on the laser/detector housing, yielding a total optical path length of 10,479 mm (10.479 m). Air flow in this additional path prevents any trapped air that could bias the water vapor measurements. The mirror airfoils were designed to maintain the air density between the two mirrors equal to the free stream density (see Section 4.2). Outside the mirror airfoil, between the $M_0$ mirror airfoil and the laser housing, the geometry of the parts forms a venturi. At typical flight air speeds, the pressure between the airfoil and the laser housing will be lower than the ambient; there will be air flow from the hole in mirror $M_0$ through the airfoils due to this differential pressure. The total absorption path length is calculated based on a ray trace, knowing the Herriott configuration for fifty passes between mirrors of known radii of curvature (Altmann et al., 1981; Herriott et al., 1964). This value is thought to be precise to one part in 10,000 over the ambient temperature range of 100 K in flight because the length of the long axis of the multi-pass cell is defined by invar spacing rods, which have a coefficient of thermal expansion of $10^{-6}$ $K^{-1}$. The TDL is mounted on a small aluminum plate, attached to a thermoelectric cooler (TEC) maintained at a specified temperature that is actively stabilized to ±0.01°C to minimize wavelength shift of the emitted infrared radiation (see Section 2.2.1).

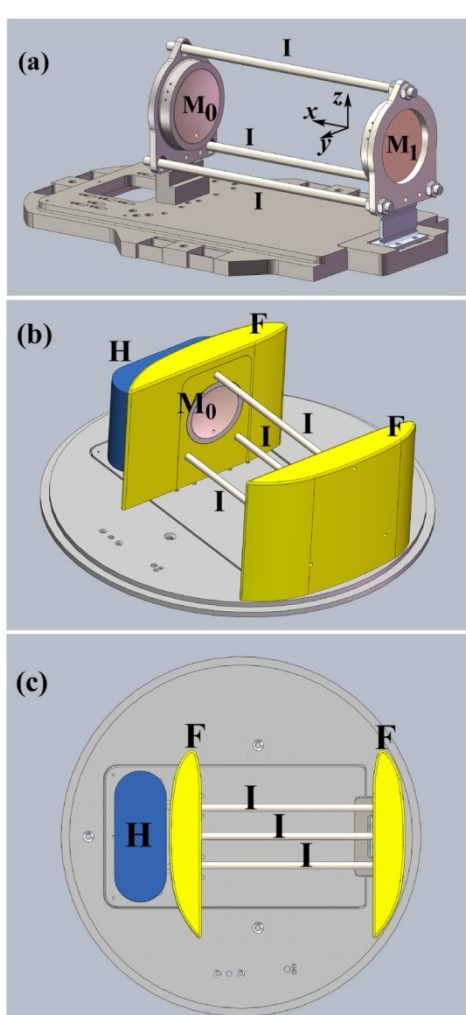

**Figure 2.** (a) CAD drawing of the JLH Mark2 Herriott cell consisting of two plano-concave mirrors, one of which ($M_0$) has a small orifice (barely visible at the bottom of $M_0$) through which the laser beam is injected at a specified direction. The beam undergoes 50 traversals between the mirrors and exits the cell through the same orifice (Section 2.1). The separation between the mirrors is fixed at a specified value by firmly attaching them to the ends of three Invar rods (I). Also shown are the x, y, z coordinates referred to in Section 4. (b) CAD drawing of the instrument showing two aerodynamic fairings (F) used for deflecting the airflow around the mirrors during flight, thus minimizing the distortion of airflow between the mirrors, as described in Section 4.2 (the direction of air flow is along y). H is an evacuated housing for the laser and the detector, and is equipped with an infrared-transmitting window. (c) CAD drawing of the top view of the instrument.

## 2.2 Instrument Electronics and Software

### 2.2.1 Basic Electronics

The JLH Mark2 electronics hardware includes a laser/detector pair for sensitive measurement of water vapor, the main JLH flight computer for laser control and signal processing, and an upgraded interface processor board (IPB) for onboard processing and network communication. The laser control and detector signal chain of JLH have been described previously in May (1998). Minor changes made to data acquisition and instrument implementation are described below in Sections 2.2.2 and 2.2.3. Figure 3 shows a block diagram of the laser control and signal processing electronics.

The laser current scan is produced as described in May (1998) using an audio codec integrated circuit in continuous direct memory access. However, in our implementation, we do not utilize the codec for translation and thus are not limited by the codec data rate but rather by the bit resolution - 16 bits - and an update rate of 256 kHz (equivalent to the 2f demodulation frequency). The TDL

is driven with a sawtooth-shape current ramp at a ramp repetition rate of 8 Hz to scan across the targeted water absorption transition. Data are acquired at 4 kHz (512 points per ramp). The measurement method switches between second harmonic wavelength modulation and direct absorption measurement (Section 5). For direct absorption measurement, water absorbance is quantified by direct transmission, with ten spectra co-added in software to save an averaged spectrum at 0.8 Hz. For harmonic measurement (wavelength modulation spectroscopy), the audio codec digital-to-analog converters generate a small-amplitude sinusoidal

waveform at frequency f = 128 kHz that is added to the sawtooth ramp. The detector signal is demodulated at frequency 2f = 256 kHz to yield the second harmonic spectrum.

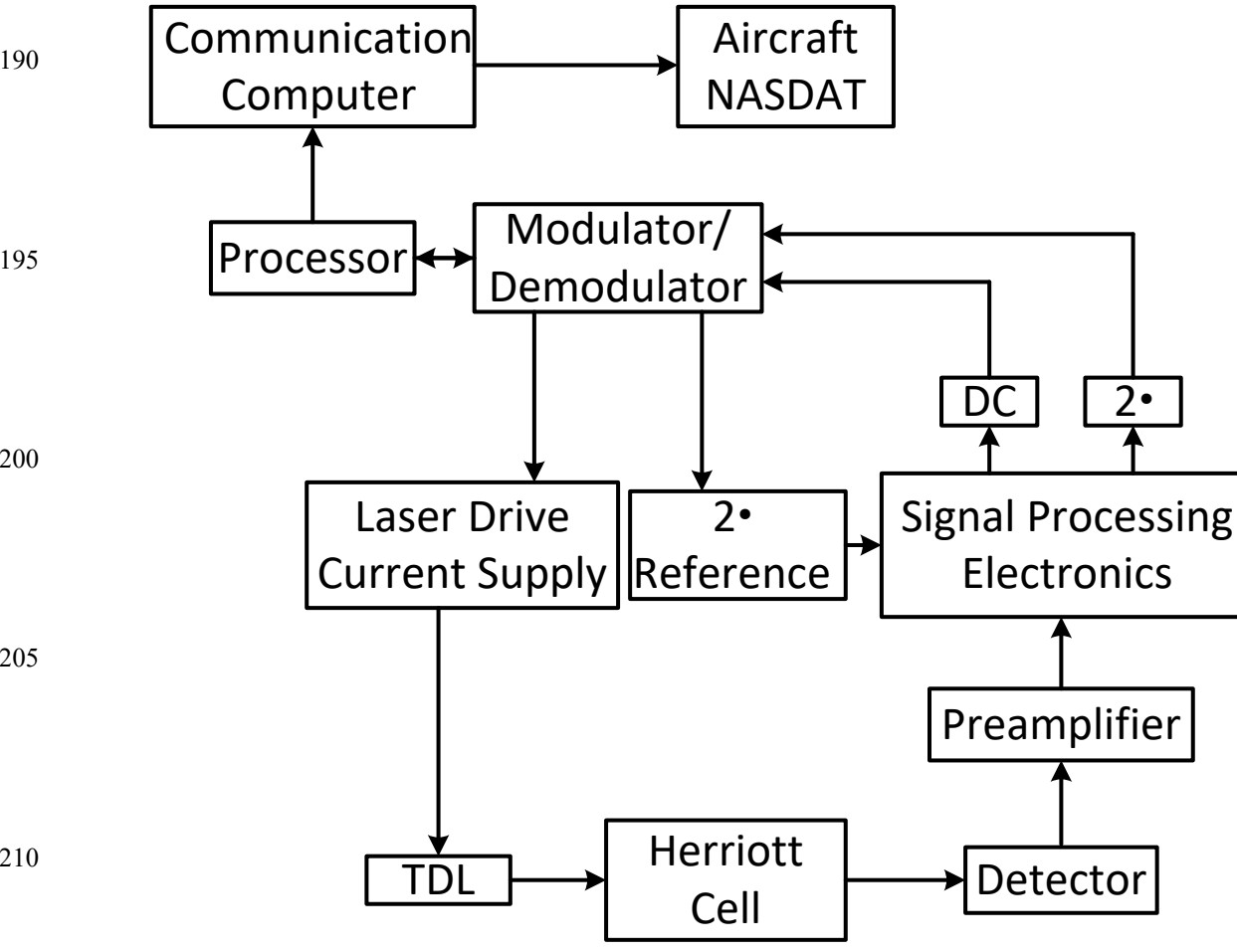

**Figure 3.** Block diagram of the electronics.

The flight computer is a single-board computer (SBC) running the MS-DOS operating system with an 80486 microprocessor. This is the legacy Ampro PC-104 motherboard from May (1998), which is an AMPRO 486D-02506 (S/N 6472175B), A60739, manufactured in 1996 and running the DOS operating system. It formerly flew on the JLH short-path instrument on the NASA

DC-8 platform in CAMEX-3 (2001). The flight computer, video card, and primary and secondary electronics boards are all in PC104 format, operating at +5 V DC and +12 V DC. Largely this PC-104 stack has preserved the electronics from the mid-1990s technology (May, 1998) for consistency in measurements. Laser drive and data acquisition are carried out using the MicroNIR Primary Board Rev. 5 (MCD 97-0200C), from 1998. This board uses the Crystal Semiconductor CS4215 chip, which led to a more

compact, low-power implementation than would have been provided by any other approach available in the mid-1990s. Temperature and engineering data are acquired by the Secondary Board PF-194V-0 (MCD 50706-B). Files are written to a flash card, legacy M-Systems DiskOnChip 2000 DIP, 128 MB, P/N MD2202-D128-X (extended temperature range), Ver. 5.1.2. Aircraft 28 V DC power is converted to +5 V and +12 V DC using Vicor VI-JW0-IY and VI-JW1-IY DC-DC converters with VI-AWW-IU input attenuation module and VI-RAM-I1 ripple attenuation module (from the mid-1990s) with additional filtering and overvoltage protection.

The electronics are temperature stabilized for greater accuracy and stability. The aluminum baseplate of the aluminum electronics box is stabilized to 298.0 K by a Minco strip heater (HK5493R6.1L24E) in pulse-width modulation control by the Secondary Board (May, 1998). The observed temperature is 298.0 +/- 0.1 K except for transient fluctuations of up to 0.5 K for 2 minutes when the aircraft rapidly changes altitude (and thus ambient temperature). The aluminum heat-sink on one side of the thermoelectric cooler (TEC) is heated to 288.0 +/- 0.1 K by a Minco strip heater (HK5163R44.0L24B) controlled by a commercial temperature controller (MPT2500, Wavelength Electronics Corp.). The measured stability of temperature is +/-0.1 K during flight when the laser housing is colder than 288 K.  For greater stability, the laser and detector on the aluminum cold block on the other side of the TEC are further temperature stabilized by active heating/cooling of the TEC. The TEC temperature is regulated by a subminiature temperature controller (HY5610, Hytek). This controller operates by proportional/integral control to stabilize the aluminum cold block to +/-0.01 K.  The HY5610 senses the resistance (corresponding to temperature) of a thermistor mounted within the aluminum cold block. This thermistor is 21 mm away from the laser housing in the cold block. This distance may affect temperature hysteresis and control (see Section 6.1).

### 2.2.2 Changes to Electronics since JLH Mark1

To improve the signal-to-noise ratio of the water vapor measurement and instrument stability, several changes have been implemented in JLH Mark2. Separate power supplies were installed for switching components (e.g., heaters and TEC) so that a dedicated low-noise power supply was used exclusively for the laser, detector, and computer. The signal chain was optimized for a narrow scan across the water absorption line at a pressure of 150 hPa (the most common flight altitude during the NASA SEAC[4]RS field mission). In the signal chain gain stages, variable resistors have been replaced by precision fixed resistors to improve temperature stability of the electronic response. In addition, all cables and boxes have been shielded for electromagnetic field (EMF) protection.

JLH Mark2 incorporates a new interface processor board (IPB) running Red Hat Linux 2.4.7-10 to augment the I/O and on-board processing capability of the JLH legacy SBC. Through trial and error, we found that adding capabilities to the CPU board interfered with the C code controlling the instrument. As a result, we added the separate IPB dedicated to "on-the-fly" processing and broadcasting the data packets. The IPB features an AMD Geode GX466 (333 MHz) processor on a PC104 form-factor CoolLiteRunner (Lippert Corp.) SBC featuring two serial ports, 10/100BaseT Ethernet and 8 GB compact flash storage (WinSystems Corp.). The IPB forms the data interface between the JLH computer and the ethernet-based aircraft data system. On the NASA ER-2 aircraft, this data system is the NASA Airborne Science Data and Telemetry System (NASDAT); Sorenson et al., 2012), and on the NGSC Flying Test Bed, the Vehicle Management Computer (VMC) collects all the data. One RS-232 serial port on the IPB is used to collect science telemetry packets transmitted by the main JLH processor. JLH packets consist of ASCII-encoded, comma separated values, terminated by linefeed character. After validating a packet receipt, values are converted from

ASCII to binary. Data are time-stamped, stored on compact flash, formatted and transmitted as NASDAT-compliant UDP packets to the NASDAT. As part of the NASDAT, the ER-2 Linkmod forwards data packets to users on the ground. The IPB also collects
pressure, temperature and standard aircraft housekeeping and navigational data in Interagency Working Group standard format number 1 (IWG1) (Webster et al., 2024) to enable on-board real-time processing of JLH water vapor data. The IPB program, written in 'C', uses multi-threading to handle the asynchronous data streams.

### 2.2.3 Instrument Software

The legacy instrument software resides on the MS-DOS computer mentioned previously. It configures the hardware, continuously
acquires the spectrometer data, and outputs a serial stream of raw data. The new communication software on the IPB then collects the serial output from the DOS computer and broadcasts it to the aircraft NASDAT or VMC. Upon receipt, the ground software processes the data "on-the-fly." We provide more details on data analysis in Section 5.

The legacy instrument software has undergone minor upgrades since May (1998). It is written in $C^{++}$ and operates as a single-
threaded application in DOS. The basic data processing technique is similar to that of May and Webster (1993). The following are the user-selectable input parameters: laser scan rate, laser modulation frequency and depth, number of scans to be averaged, number of scans before the data are written to the flash drive, and the heater parameters. At the start of the program execution, the hardware, heaters and laser scan are initialized. In the next step, an infinite loop is initiated and data are acquired. In the timing sequence, shown in Figure 4, the TDL is first off for 3 ms to acquire the "null-point." The laser scan ramp with a duration of 125 ms then
turns the laser on and tunes the laser across wavelength region of interest. During the ramping, the laser is also modulated at a frequency f = 128 kHz which is superimposed on the current ramp (Figure 4). The time-varying signal is detected at frequency 2f = 256 kHz and filtered to remove DC offsets caused by the slow laser current ramp. The peak-to-peak amplitude of the 2f spectrum is saved to disk along with laser power and other engineering parameters. After the ramp, the laser is turned off again for 3 ms to acquire a "null" signal. This on-off sequence is then repeated. The results from individual wavelength scans are averaged
continuously by the software. As a compromise between the instrument precision and time resolution, 10 scans are averaged and mean data are reported every 1.28 s. Every ~20 s, the laser modulation is turned off and a direct absorption signal is acquired (again a 10-scan average).

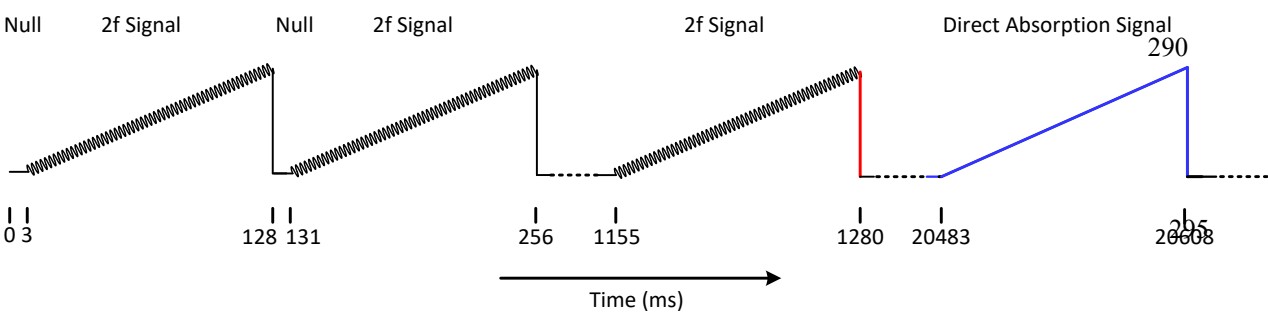


**Figure 4.** The data acquisition timing sequence (schematic, not to scale). Between t=0 and 3 ms, the scanning ramp is off and a "null signal" is acquired. At t= 3 ms, a ramp of 125 ms duration with a superimposed modulation is applied and the 2f signal is acquired (the modulation depth is exaggerated for clarity). The on-off sequence is repeated ten times until t=1280 ms, at which point (shown in red) ten 2f signals are averaged and a mean is reported. After the 160[th] cycle, at t~20 s, the laser modulation is turned off (shown in blue) and a direct absorption signal is acquired
(again a 10-scan average, not shown here).

The new communication software transmits real-time data to the aircraft data system (Section 2.2.2). The ground software processes the broadcasted data stream from JLH (spectroscopic data) and pressure and temperature (provided by MMS on the NASA ER-2, and facility sensors on the FTB). Water mixing ratios are calculated using the algorithms of May and Webster (1993).

Measured peak-to-peak 2f (pp2f) amplitude is compared with a look-up table of expected pp2f as a function of pressure and temperature. The expected pp2f is interpolated between the table values of the nearest pressure and temperature to calculate the mixing ratio of water (May, 1998).

## 2.3 Calibration

### 2.3.1 Laser Characterization

In this section, we discuss the absorption line selection and laser characterization. The JLH Mark1 instrument originally used a laser that sampled the strong 7294.1229 cm$^{-1}$ water absorption line. This line is flanked by two weak water lines, thus slightly complicating the analysis. What is new for JLH Mark2 in SEAC$^4$RS and FTB is a new laser that samples the strong water line at 7299.4311 cm$^{-1}$. This line has simpler data processing because it is isolated from other water lines.

Laser characterization is necessary for accurate water vapor measurements. We first determine the absolute laser wavelength to within ~0.18 nm (1 cm$^{-1}$) with a Burleigh wavemeter. The wavelength and laser tuning rate with wavenumber are refined by using seven identified spectral lines of CH$_4$ as wavelength standards in the same spectral region as the water line of interest (Brown et al., 2013; Rothman et al., 2013). The CH$_4$ line positions are known to within 10$^{-4}$ cm$^{-1}$ (Brown et al., 2013), allowing the accurate determination of the water line position by interpolation. The laser tuning rate here refers to wavenumber (cm$^{-1}$) versus index in

the laser scan, which goes from 1 to 512. For this purpose, measurements are made with JLH Mark2 in a laboratory test chamber. These weak CH$_4$ lines are measurable by filling our calibration chamber with 27.6 hPa pure CH$_4$ (with trace water). We fit a second-order polynomial to wavenumber versus index. Ultimately this wavenumber scale is needed to fit the direct absorption using fundamental physical information.

Other laser parameters of interest are spectral purity, linewidth, and laser modulation amplitude. Spectral purity means the fraction of laser emission centered on a single wavelength and not in side bands (e.g., at a shifted wavelength). The measurement of water mixing ratio is directly proportional to the spectral purity. The spectral purity of the laser is verified by measuring the laser transmission through a sample at very high water vapor concentrations, which absorbs all the light at the line center wavenumber of 7299.4311 cm$^{-1}$. The laser linewidth affects the amplitude of the pp2f signal and fitting of direct absorption, so it is important

to characterize. The laser linewidth is determined in the laboratory by measuring the water absorption lineshape at low pressures where the absorption linewidth is solely due to the Doppler width. If the line is wider than the predicted Doppler width, it is assumed to be a convolution of the laser linewidth and Doppler width (Section 5).

The laser modulation amplitude is optimized for maximum second harmonic peak-to-peak signal from a water line that is pressure-

broadened at 150 hPa total pressure. The modulation amplitude is set by the user in software in electrical units (mA current to drive the laser). This amplitude is not an intrinsic laser property, but the relation between laser current and output wavelength is. It is critical for water vapor measurement accuracy to know the actual modulation amplitude in spectroscopic units (wavenumbers, cm$^{-1}$) because the expected pp2f amplitude at different pressures and temperatures in the look-up tables is sensitive to the modulation amplitude. This is determined in the laboratory by placing JLH Mark2 inside a continuous flow chamber (Section 2.3.2), measuring

well-characterized water mixing ratios over a range of pressures from 100 hPa to 700 hPa, and comparing the JLH Mark2 signal to synthetic modulated spectra. This method of estimating laser modulation is described by May and Webster (1993).

### 2.3.2 Spectrometer Calibration

The instrumental parameters of interest are the optical path length in the multipass absorption cell and the instrument response (sensitivity). The optical path length is checked in the laboratory by individually identifying each pass of the laser light between the mirrors with an infrared detection card, covering and uncovering each spot on the near and far mirrors. The fraction of light impinging on the detector from the final reflected pass (as opposed to an adjacent pass) has been measured to be greater than 99.9%. At large optical depths (e.g., measurements on the ground in humid air), no contribution from scattered light was detected.

The instrument response is characterized through continuous flow of an accurately known air/water vapor mixture through the JLH sample cell when it is inside the laboratory calibration chamber (see below). JLH water vapor measurements are calculated as in the field data. The ratio of known source water data to JLH water data is used as the instrument response factor and is used to scale the field measurements.

By 2007, improved calibration methods were developed, which allowed for laboratory measurements of sample atmospheres in test chambers with water volume mixing ratios as low as 2 parts-per-million by volume (ppmV) (Troy, 2007). The source of the air-water mixture for determining the instrument response mentioned above is a commercial Thunder Scientific 3900 (TS3900) low-humidity generator (https://www.thunderscientific.com/), which can provide air with stable water mixing ratios from less than 1 ppmV to 12,000 ppmV. The carrier gas is ultrazero air from cylinders. The TS3900 uses a technique that generates air with a constant water mixing ratio to an accuracy of 0.1 K frostpoint (corresponding to better than 1% accuracy of mixing ratio). The technique is a primary standard recognized by NIST and used by commercial hygrometer manufacturers to check their instruments. It depends solely on accurate measurements of pressure and temperature to achieve a constant mixing ratio of water vapor in air flowing over a saturator. By scaling the JLH data to the TS3900, the standard we are using to calibrate the instrument is the TS3900.

For an open-path instrument such as JLH, it is necessary to place the laser/detector housing and sample cell inside a test chamber. The JLH test chamber used for laboratory calibration is a 304 Corrosion Resistant Steel (CRES) box vacuum chamber manufactured by the Kurt J. Lesker Company, and coated on the inside with a hydrophobic coating of Fluoropel® 1302IBA, manufactured by Cytonix Inc. of Beltsville, Maryland (Troy, 2007). For calibration of the JLH instrument in the lab, an air/water vapor mixture generated by the TS3900 flows continuously through the CRES chamber at constant pressure and temperature. The sampled air mixture is checked both upstream and downstream of JLH with a reference Vaisala DM500X precision surface-acoustic wave (SAW) hygrometer. Typically, half of the flow is directed through JLH and the rest through the Vaisala hygrometer, which has a quoted accuracy of 0.3 K frostpoint (corresponding to 2% accuracy of water mixing ratio).

Below, we describe a recent example of JLH laboratory calibration. Figures 5a and 5b show time series of JLH measurements within the lab calibration chamber as 200 ppmV and 50 ppmV water vapor, respectively, are flowed through the chamber.

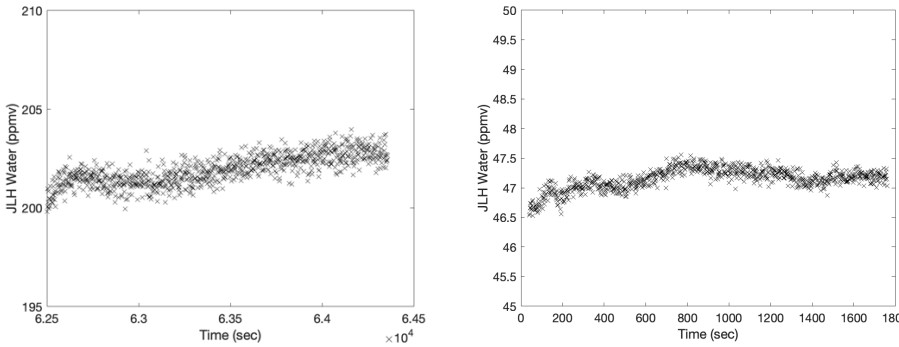

**Figure 5.** Time series of JLH water vapor measurement inside the laboratory calibration chamber with continuous flow of air/water vapor mixture from the TS3900 humidity generator. (a) Left plot is 200 ppmV input at 57.2 hPa total pressure and 295.4 K temperature, (b) Right plot is 50 ppmV input at 120.3 hPa total pressure and 296.4 K temperature.

Figure 5c demonstrates the linearity of JLH response versus reference gas standard from the TS3900 from 3 ppmV to 200 ppmV. This is also shown in Table 3 along with the difference in percent and ppmV. Generally, results are within 9% but there is a pressure dependence as shown in Fig. 5d. This pressure dependence has been documented in our previous laboratory calibrations. At higher and lower pressures, the response changes systematically.

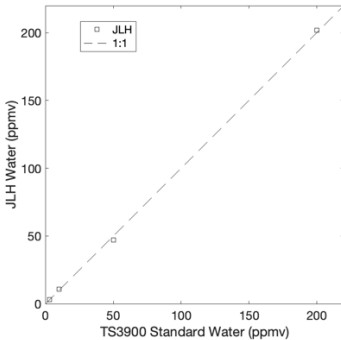 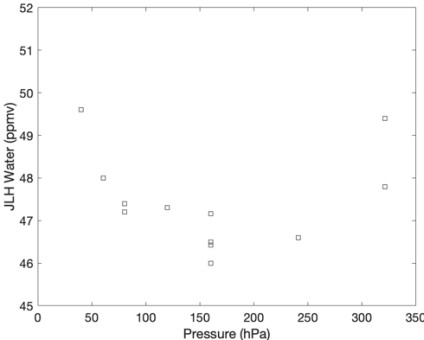

**Figure 5.** (c) Left plot shows the relation of JLH water measurement to TS3900 water source. A 1:1 line (dashed line) is also shown. (d) Right plot is JLH water measurement inside lab calibration chamber at different total pressures while the TS3900 provides a constant source of 50 ppmV water vapor at room temperature.

The best estimate of JLH total error for NASA SEAC[4]RS and more recent FTB flights is 12%, adding in quadrature the following terms. The largest error source is 9% in water measurement due to mismatch of the actual lineshape and the predicted lineshape as manifested as pressure-dependence (Fig. 5d). We estimate 5% uncertainty in water measurement from uncertainty in the temperature dependence of the actual lineshape and 6% from scaling to laboratory calibrations (Fig. 5c and Table 3). There is 1% error in water measurement due to uncertainty in the in-flight temperature and pressure measurements. Finally, only 0.1% uncertainty in water measurement is due to uncertainty in the path length, and a fraction of a percent from other error sources such as wavelength, laser spectral purity, light scattering, et cetera.

## 2.4 Instrument mounting on aircraft platforms

Over the years, the JLH instruments have been mounted on various locations on scientific aircraft. Examples include lower camera ports underneath the NASA ER-2 wing pod (May, 1998) and ER-2 Q-bay (Figure 1a), hatches underneath the wing of the NASA WB-57F aircraft, and most recently the fuselage of the Northrop Grumman Systems Corporation FTB (Figure 1b). The airflow is the free-stream flow across the instrument between the mirrors, which are farther away from the aircraft than the estimated boundary layer. There is no sample inlet or outlet in the conventional sense and the sample is constantly flowing across the instrument axis. The sample replenishment time is estimated to be ~0.3 millisecond across the optical path, assuming an air speed of 200 m/sec. Computational fluid dynamics calculations for the WB-57 aircraft indicate that the pressure and temperature at the mounting location are not significantly different from ambient values, within the uncertainties of the in situ static pressure and static temperature measurements (Engblom, 2003). The open-path cell and its location on the aircraft ensure accurate water measurements without wall-effect contamination, especially in ice supersaturated air and contrails.

## 3. Design issues in JLH Mark1

### 3.1 Optomechanical stability

The original JLH Mark1 instrument was designed and built in the early 1990s, and flew for over a decade in its original configuration (May, 1998). With the accumulation of years of flight data, it became clear that the JLH instrument needed redesign and upgrades due to some inherent design weaknesses, and also due to wear and deterioration of the instrument from years of use.

Design weaknesses in the original JLH resulted in thermal instability of the optics over the wide range of temperatures (180 to 300 K) experienced during airborne field missions. This caused loss of light intensity at the detector as the instrument temperature decreased with altitude. Due to temperature induced distortions of the optical bench and mirror holders, the laser beam path would shift until a portion of the beam would not impinge upon the detector. Because the laser / detector system measures the ratio of laser power while tuning across the water absorption line to the laser power off-line, the instrument is self-normalizing to a certain extent. However, with signal strength loss due to beam drift and reduced mirror reflectivity due to wear and deterioration, the signal-to-noise ratio would decrease. This eventually became a limiting issue in the NASA MACPEX campaign in 2011 (see Table 1).

A major problem with the original optical bench design was that it was structurally over constrained. The optical bench was constructed from materials that had markedly different coefficients of thermal expansion (CTE). As the instrument temperature decreased with altitude, the materials contracted by different amounts, pressed against each other, and induced high stresses. This caused distortion of the optical bench. For example, the original JLH design for the Herriott cell consisted of two spherical mirror holders rigidly held by three Invar rods. This Herriott cell assembly was rigidly bolted to a flat plate made of G-10 fiberglass. As the temperature of this system decreased, the Invar rods experienced very little contraction, whereas the G-10 plate shrank by almost two orders of magnitude more than the Invar rods. This caused the entire optical bench to distort and impressed forces of several hundreds of Newtons onto the Herriott cell. The solution to this problem was to redesign the optical bench so that all components were kinematically mounted (Section 4.1).

## 4. Improvements implemented in JLH Mark2

### 4.1 Optomechanical redesign

For the new optomechanical design of JLH Mark2, we used kinematic mounting to improve instrument stability. In kinematic mounting, an instrument, or an element of an instrument, is mounted such that each of its six degrees of freedom (DOF) is constrained non-redundantly by the mountings; there are only six physical constraints that uniquely control the six DOF. True kinematic mounting prevents any forces from being induced between any of the mounted components. Approaching ideal kinematic mounting becomes very difficult and expensive in real physical systems due to non-ideal components experiencing distortion and friction at very small scales due to applied forces. However, there are methods of design by which the induced mounting forces can be minimized at the constraints such that effect upon the mounted system is negligible. This is referred to as quasi-kinematic, or semi-kinematic, mounting. In engineering documentation, when a mounting system is referred to as kinematic, it is usually a reference to a quasi-kinematic mounting system; we use that meaning for kinematic here. For a more detailed exposition of these principles, see Yoder (2008).

To address the stability issues noted in Section 3.1, a combination of materials selection and kinematic design were used. The new design has the $M_0$ mirror in the Herriott cell mounted rigidly to a steel plate, as shown in Figure 2. The change was made from G-10 fiberglass to steel to increase strength and resistance of flexure of the system due to vibration. 15-5 PH steel was used due to its low CTE; the CTE is on the order of 60% of many other steels. The $M_0$ mount constrains motion of the Herriott cell in all six DOF. The $M_1$ mirror is attached to the steel plate using a spring mounting, or flexure. This flexure constrains the $M_1$ mirror in five DOF.

The $M_1$ mirror is allowed to move along the x axis, resisted by a very low spring force. If the $M_1$ mirror were also rigidly mounted, the constraint of the x degree of freedom would be redundant, and forces would arise between the mirrors as the temperature of the instrument changed, which would cause distortions in the system. Instead, because of the flexure mount, as the temperature decreases and the steel plate contracts, the $M_1$ mount flexes in the x direction and a force of only 22 Newtons is applied along the Invar rods between the two mirror holders. The force is not zero, but it is two orders of magnitude less than in the original, non-kinematically, mounted JLH Mark1 Herriott cell.

The steel optical bench plate is mounted to an aluminum adapter plate, which is, in turn, mounted to an aircraft structure. The adapter plate is used so that JLH Mark2 may be mounted on multiple airframe structures. As most aircraft structures are made of aluminum, there is a CTE match and the adapter plate can be bolted directly to the aircraft with no need for flexural mounts. Because of the difference in CTE between the steel and aluminum, the optical bench itself is mounted to the aircraft adapter plate using three flexures.

Within the Herriott cell, the mirror holders, which are made of Invar, are connected by three screws to the mirror mounts, which are made of titanium. Two of the three screws clamp the mirror holders to small flexures in the titanium mounts. This prevents stresses arising between the mounts and the mirror holders across the span where the two materials of dissimilar CTE are clamped together. The mirrors, which are made of Zerodur (a glass-ceramic with very low CTE) are held in the mirror holders by three pads of high strength silicone adhesive. The silicone pads act as flexures that prevent buildup of stresses between the mirrors and the mirror holders.

## 4.2 Aerodynamic fairings

The authors considered aerodynamics in the redesign of JLH Mark2 to minimize distortion of the airflow in the optical path of the external Herriott Cell, and to improve stability. The new design features aerodynamic fairings resembling stubby wings that maintain a constant air density between the mirrors. In JLH Mark1, these fairings had the form of symmetric ellipsoidal airfoils (May, 1998). That Mark1 configuration with a pair of airfoils would create a venturi effect, thus accelerating the flow between the two fairings. The associated pressure reduction is expected to cause a reduction in air density precisely where the instrument measures absolute water content. The final specific moisture content would thus be erroneous because it is computed using the free-stream value of air density. In addition, the JLH Mark1 airfoils experience significant deflection during flight.

In the modified JLH Mark2, we designed new fairings to overcome the above issues. We reshaped the fairings using cambered airfoils to keep the air density between the two fairings equal to the free stream density, which is equivalent to having a pressure coefficient of zero in the region between the two fairings. This also implies that all of the air approaching the leading edge of the fairings needed to be deflected around the top surface of the airfoil. Here, we are using "top" and "bottom" in relation to the usual orientation of wings on airplanes. These fairings are not horizontal when mounted as part of the instrument. It was reasoned that since these fairings do not disturb the flow going underneath them, two identical such fairings could be placed with their bottoms facing one another and that the flow around each would behave independently of the other. In other words, two identical airfoils could be used and the flow around them would not interact. This simplified the aerodynamic analysis by not having to treat the two wings as being part of a biplane. This lack of interaction between the two fairings has not been proven, but it was simply assumed to be true and the airfoil shape was then developed in isolation.

There were some constraints on the shape of the airfoil besides the goal of achieving a pressure coefficient close to zero on the lower surface. The physical mounting required the instrument to fit through a hole in the surface of the aircraft, the 50.8 cm diameter camera port in the lower Q-bay of the ER-2 aircraft. That space constraint limited the extent of the airfoil upper surface. This mounting constraint also prevented a long tapered trailing edge, resulting in a rather steep aft surface. The fairings, of course, had to provide sufficient room inside them for the Herriott cell mirrors that they are intended to protect. They also had to

accommodate a small amount of adjustability in the spacing between the two mirrors. In order to simplify machining, alignment and mounting of the two fairings, we kept a large portion of the bottom surface flat (similar to the Clark Y airfoil). A similar self-imposed constraint was to construct the shape of the airfoil using only circular arcs and keeping them tangent at the transition points from one arc to the next. The pressure distribution on the airfoil was computed using XFOIL, a powerful, free, two-dimensional flow solver developed by M. Drela at MIT (Drela, 2013). The final shape developed here was the result of only four

iterations. Treating the airfoils as two-dimensional means that we implicitly neglected any effects due to trailing wingtip vortices. Given the low aspect ratio of these stubby wings, there must be some spanwise flow towards the tips in the region between the two fairings. We did not make any attempt to quantify the impact of this effect. Figure 6 shows the final shape of the fairing and Figure 2 shows the 3-dimensional CAD drawing of the fairings in the JLH Mark2 instrument.

In addition to the aerodynamic improvements, the JLH Mark2 design isolated the Herriott cell system from the airflow management system of the fairings. The JLH Mark1 design had the mirrors mounted directly to the airfoils, which were, in turn, mounted directly to the optical bench plate. In addition to the cell distortion issues due to the structural over constraints, deflections due to aerodynamic loads on the airfoils affected the mirrors. The JLH Mark2 design eliminated this issue by making the Herriott cell and the fairings independent systems. The Herriott cell and the fairings are both mounted to the optical bench, but there is no

physical contact between them. In this way, the Herriott cell is isolated from any distortions and vibrations of the fairings due to aerodynamic loads.


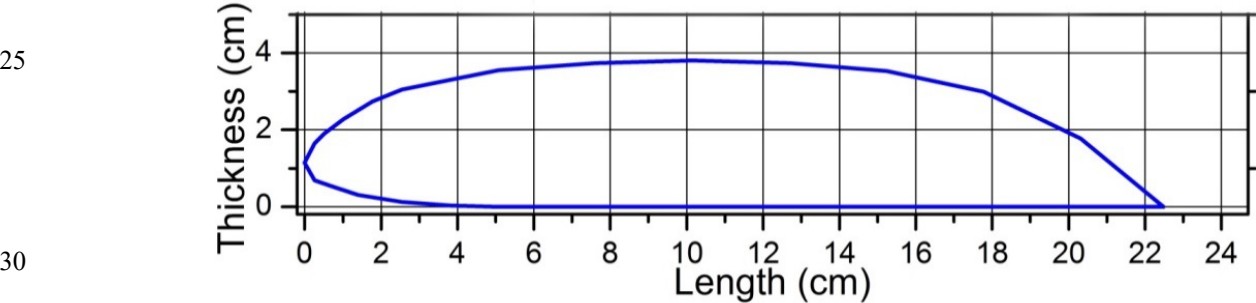


**Figure 6.** Airfoil shape for zero coefficient of pressure on much of the lower surface.


## 5. Data Analysis

The data reduction technique follows closely on May (1998) and May and Webster (1993). The accuracy of the water concentration obtained from JLH Mark2 measurements is limited primarily by uncertainties in the laser parameters and spectral fitting. In the section below we describe the steps to measuring the parameters needed for accurate water vapor measurement.

As discussed in Section 2.3.1, laser tuning rate (wavenumber versus index point) is determined by fitting seven $CH_4$ lines surrounding the targeted water line with a second-order polynomial. The instrument line shape is then characterized by comparison of a low-pressure absorption feature with the expected profile of a Doppler line shape. Figure 7 shows an experimental line shape obtained with the Herriott cell in a laboratory test chamber at a temperature of 297.54 K with pure water vapor at a pressure of

$9x10^{-2}$ hPa. The observed absorptance, $(I-I_o)/I$, is only ~0.134, so the sample is optically thin and the line shape is due to only Doppler broadening. This Gaussian is 20% broader than the calculated Doppler-broadened lineshape for this measured temperature. We hypothesize that the difference between the Gaussian fit and the calculated Doppler broadening is dominated by fluctuations of the laser wavelength (associated with either temperature or dithering noise), but have not fully diagnosed the issue.

JLH Mark2 uses a combination of fast 2f spectra and less-frequent direct absorption spectra to calculate water vapor mixing ratios. As described in Section 2.3.3, 2f spectra are acquired at 8 Hz, and averages of 10 harmonic spectra are written to the flight computer for real-time communication to the ground and subsequent analysis. Figure 8 shows one such raw second-harmonic (2f) spectrum (black trace) in the stratosphere during a flight on 2 September 2013. The blue trace is the return laser power at the detector. At index 44, the laser power drops to its zero level (with a very small contribution from background light) during the 3-msec laser off

period, described in Section 2.2.3. The laser power increases with applied current as the index increases. We note that this is a stratospheric measurement with less than 10 ppmV water vapor. At such low concentrations, direct absorption signals are very weak and the observed 2f signal shown here is a testament to the high detection sensitivity possible in wavelength modulation spectroscopy.

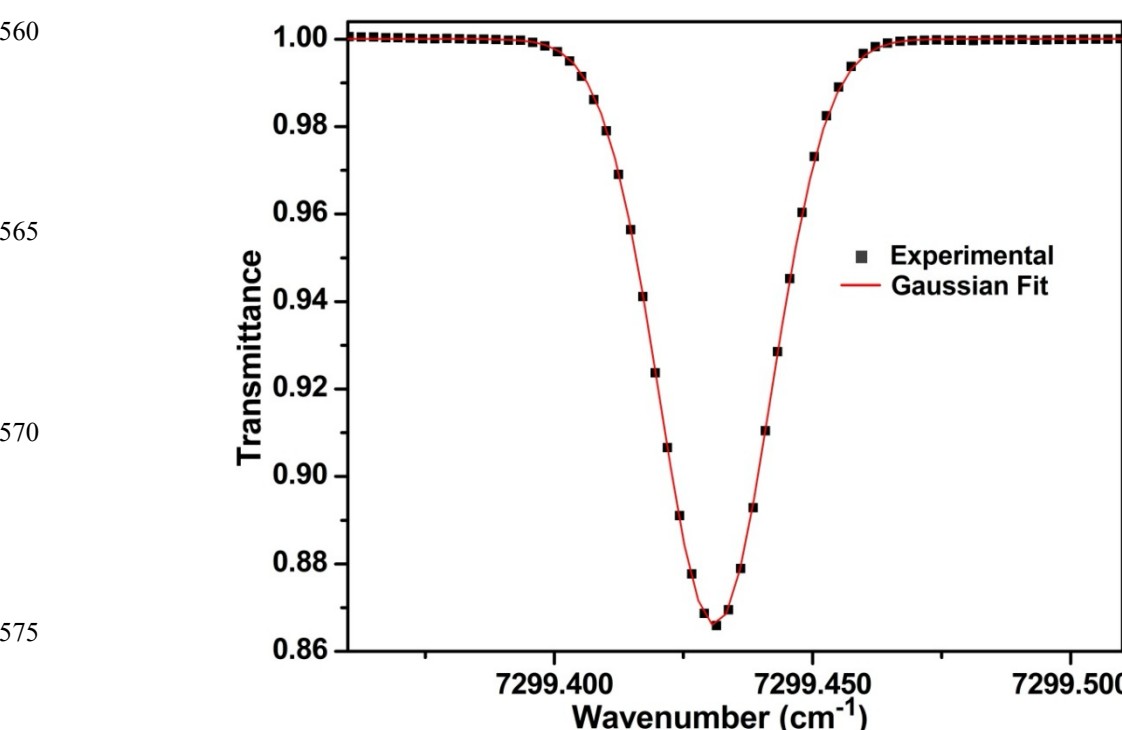

**Figure 7.** Experimental transmittance spectrum and a Gaussian fit to the data, where the transmittance = 1- absorptance.

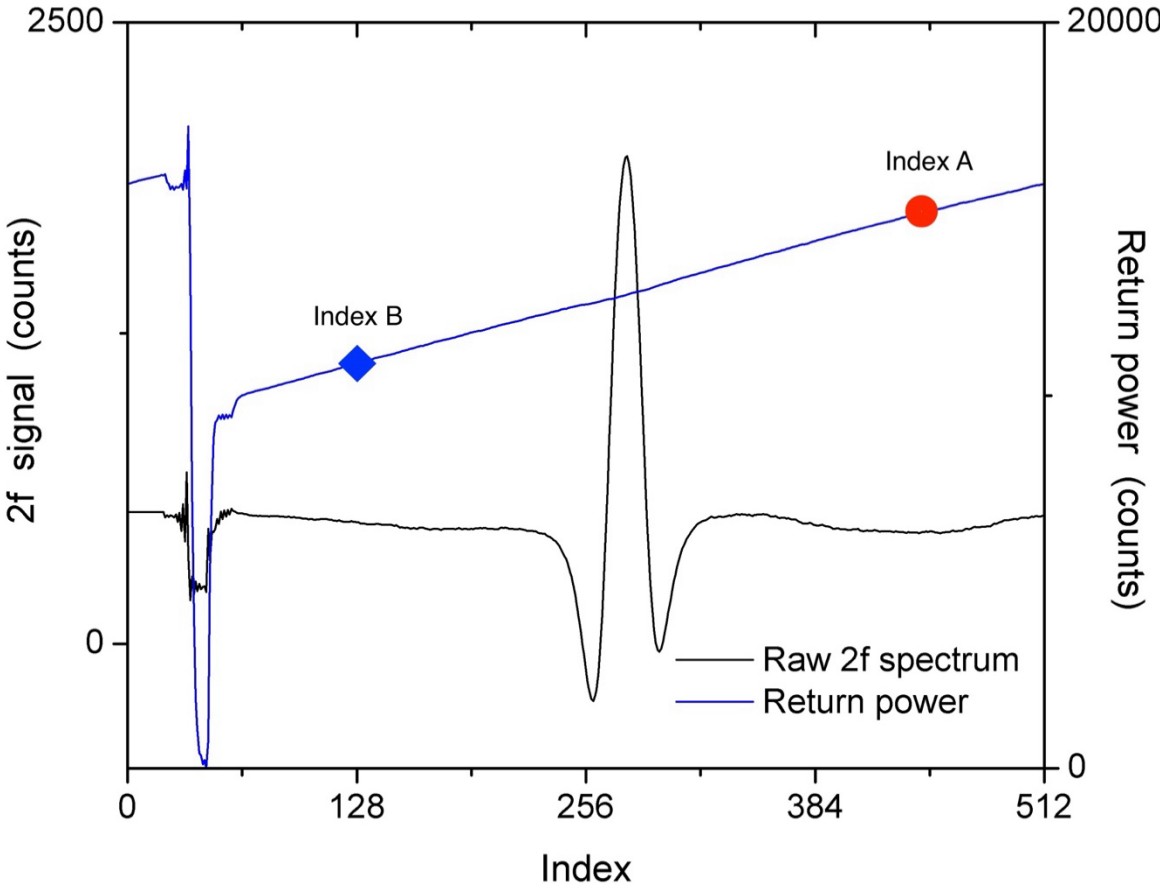

**Figure 8.** An example second harmonic (2f) spectrum (black trace) and return laser power spectrum (blue trace) acquired with JLH Mark2 during the NASA ER-2 aircraft flight of 2 September 2013. The spectrum was recorded at 82178 sec UTC at a pressure of 129.96 hPa, temperature of 203.3 K and a calculated water volume mixing ratio of 6.81 ppmV. As reference for Figure 10, the return laser power is identified at two points, the blue diamond is Index B and the red circle is Index A.


The 2f signal has a characteristic second-derivative type shape with two negative lobes and one positive lobe at the line center. It

has commonly been observed in the tunable diode laser (TDL) community that 2f spectra have asymmetric negative lobes, and other researchers have found reasons other than electronic filtering that can cause this artifact (e.g., Goldenstein et al., 2014). For JLH Mark2 measurements, the peak-to-peak 2f amplitude (pp2f) is measured between the first negative lobe and the positive lobe. This measurement is precise, but not as accurate as direct absorption. Fig. 7 shows curved 2f baseline due to a Fabry-Perot interference fringe from a short back-reflection in the optical system. The fringe width is greater than the width of the water

absorption line. Since JLH response is tied to the laboratory measurements, this fringe does not contribute a significant source of uncertainty to calculated water vapor at low mixing ratios.

At higher water vapor concentrations, direct absorption measurements are more accurate. Every 28 seconds, the laser modulation is turned off to record a direct absorption spectrum (ten-scan average) as mentioned in Section 2.3.3. An example of such a spectrum

is shown in Figure 9, acquired during a NASA ER-2 aircraft flight of 2 September 2013. Also shown is a fitted Voigt lineshape.

For the absorption shown in Figure 9, our analysis yields a water vapor concentration of 296 ppmV. In our analysis, we obtain the water concentration by comparing the observed line-shape with that calculated by using the spectroscopic parameters in the HITRAN database. Laboratory calibration with a reference source is then used to verify this approach.

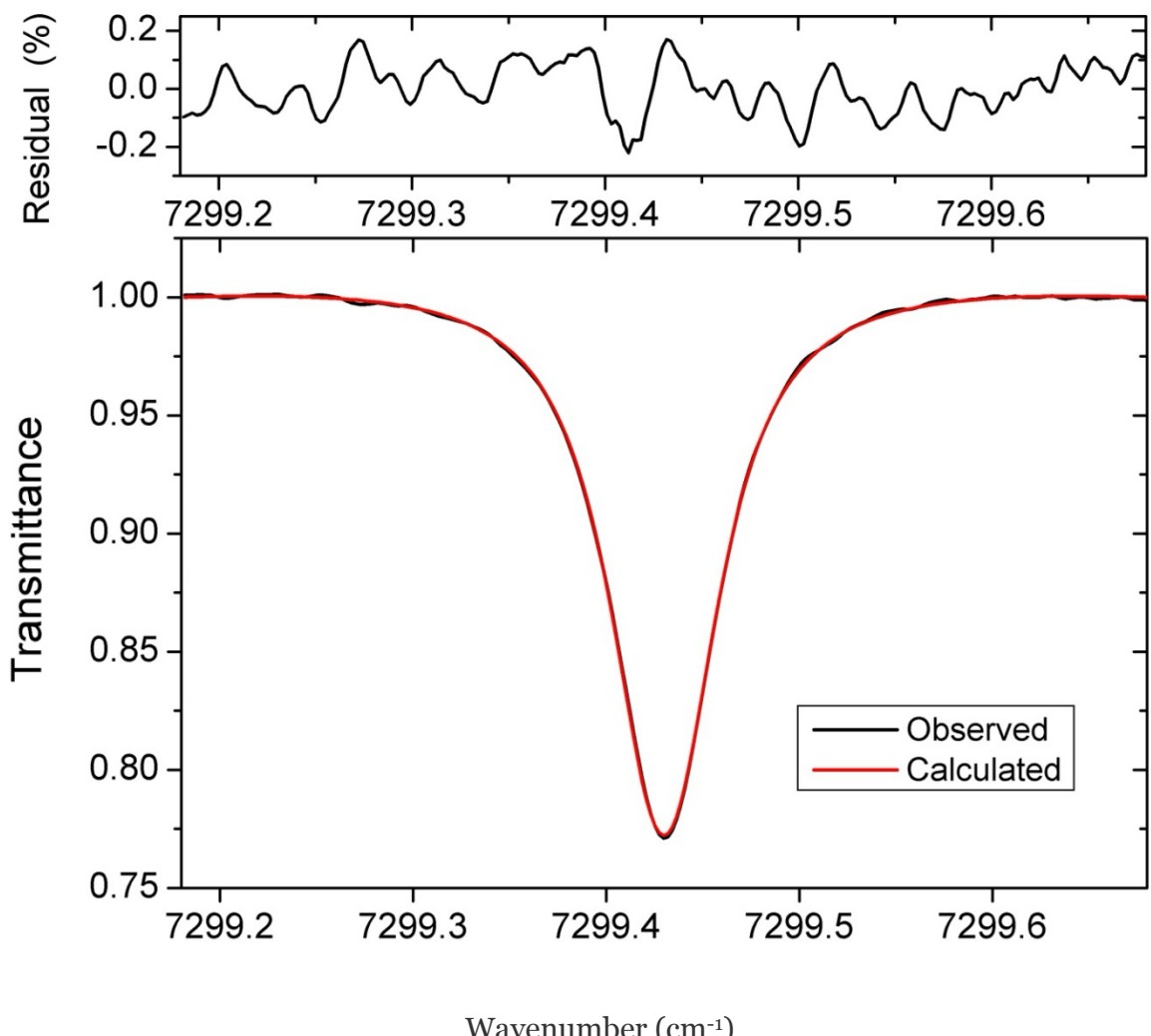


Wavenumber (cm$^{-1}$)

**Figure 9.** A high ambient pressure transmission spectrum during a flight on Sept 2, 2013, at a pressure of 225.95 hPa (169.48 Torr) and a temperature of 225.19 K. Also shown is a fitted spectrum, yielding a water concentration of 296 parts-per-million by volume (ppmV).


The residuals in the upper panel in Figure 9 demonstrate that the fits are generally good and that spectral fitting residuals are generally within JLH noise level. However, we caution that there are errors originating from the assumed line-shape function and the line parameters, often contributing to retrieval errors. The Voigt function assumed for air-broadened lines in our spectral analysis programs can lead to error on the order of 2%.

**6. Performance**

**6.1 Measured improvement in precision and optical stability**

JLH Mark2 has significantly steadier return laser power than previous optical configurations of JLH Mark1 (Mark1a and 1b). To prepare for airborne measurements, we tested JLH Mark2 in a low-temperature chamber (Cincinnati Sub-Zero Z-Plus

Temperature/Humidity Chamber) at temperatures as low as 203 K and verified stable optics. The JLH Mark2 instrument then flew

as part of NASA SEAC⁴RS and the NGSC FTB flights. Example data tracking return laser power during a single FTB flight in the

UTLS are shown in Figure 10. Indices A and B (as shown in Figure 8) were selected to be in non-absorbing regions on either side

of the water feature and thus reflect only return laser power at a given injection current. Note that the DC gain was deliberately

reduced between SEAC⁴RS (Figure 8) and FTB (Figure 10), so the signal from return laser power is proportionately lower on FTB

flights. Figure 10 illustrates that, over the static temperature range of 197 – 233 K, return laser power varies only 0.7% and 1.1%

for indices A and B, respectively. Matching trends between the two indices shows that power variation is constant across a given

laser scan, which is easily accounted for in the signal processing. Given that laser alignment and return power is optimized at room

temperature, the observed increase in power with decreasing temperature suggests that misalignment and optical losses are not

dominant and are no longer a limiting factor in instrument performance. Instead, thermal drift of the laser is the dominant factor

where lower laser temperatures result in increased output power at a given injection current. The laser wavelength is drifting during

flight because the temperature of the laser substrate is changing with ambient temperature - even for a fixed set point of the TEC

control. Line locking would solve the issues but was not implemented in this compact system.

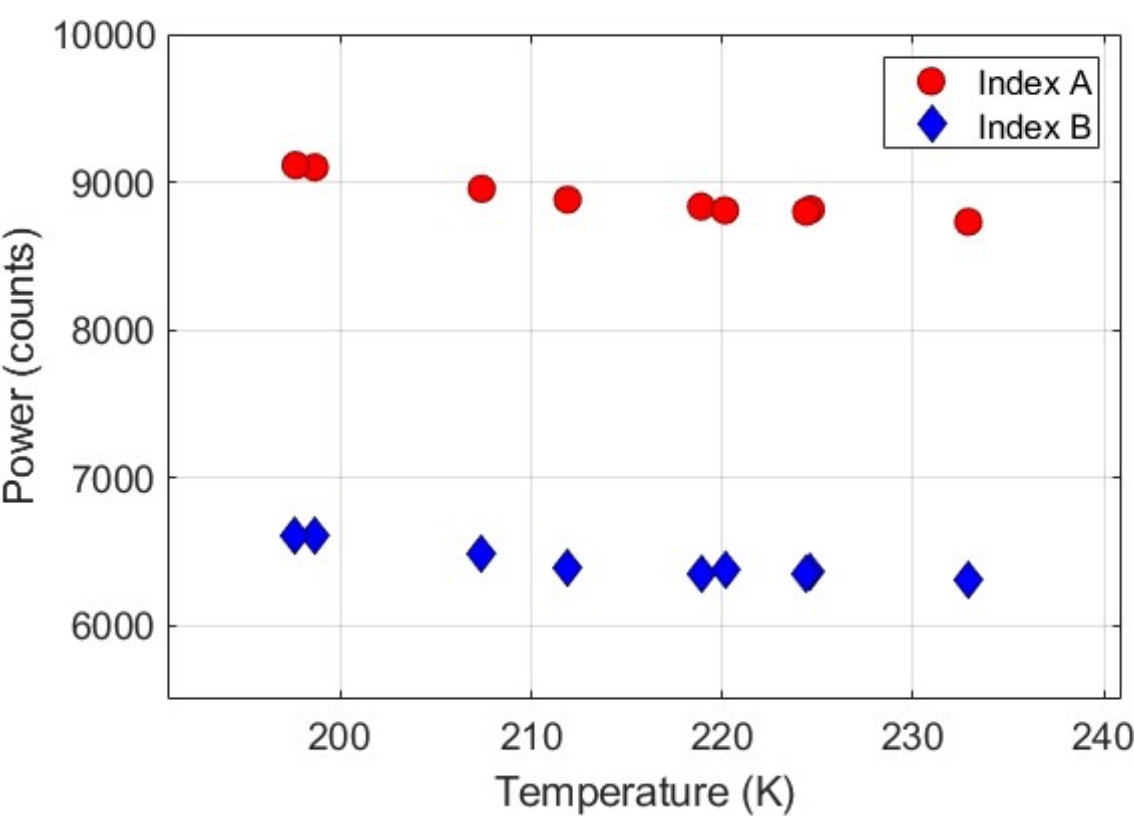

**Figure 10.** The redesigned optical mount maintains stable return laser power over the entire range of UTLS temperatures (190-240 K). Index A
(red circles) and Index B (blue diamonds) are the same as in Figure 7. Data from NGSC FTB flight of January 13, 2021.

The precision of JLH Mark2 is also better than JLH Mark1a and 1b. May (1998) reported 0.050 ppmV precision for water

concentration measured by JLH Mark1a. In practice, the JLH Mark1 precision deteriorated from year to year as the mirror

reflectivity decreased due to environmental degradation. For comparison, we show the performance of JLH Mark2 during the

SEAC⁴RS field mission in Figure 11. The stratospheric precision of JLH Mark2 was 0.015 ppmV, which is equivalent to 0.4% of

the water volume mixing ratio (3.82 ppmV).

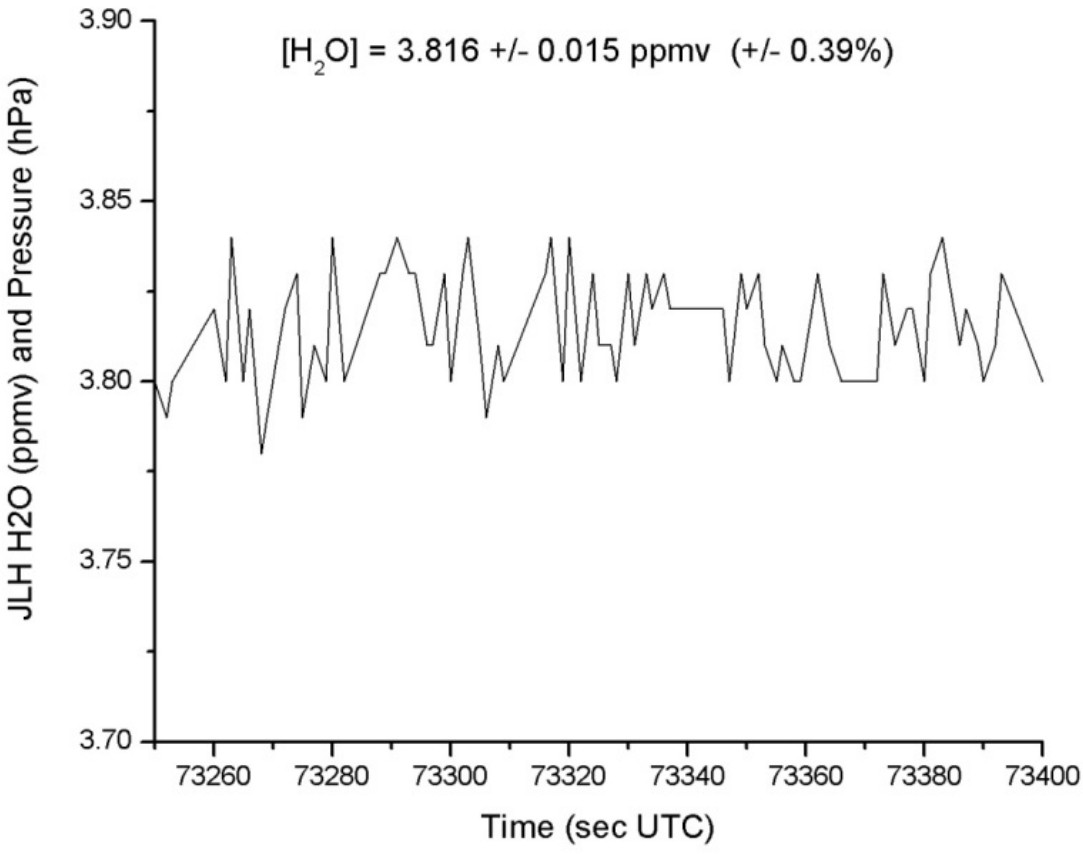

**Figure 11.** Stratospheric water measurements during level flight of the NASA ER-2 aircraft on 8 Aug 2013 recorded by JLH Mark2. In this flight segment, water vapor is expected to be nearly constant. The standard deviation of the measurement is 0.015 ppmV.


## 6.2 Comparison with other instruments

In 2007, JLH Mark1 participated in the AquaVIT-1 laboratory intercomparison of hygrometers (Fahey et al., 2014). For this multiweek campaign, the AIDA chamber was utilized as a uniformly mixed reservoir of constant water vapor mixing ratio over a range of pressures and temperatures. The core of the JLH instrument (multipass cell and laser/detector) were installed within the

AIDA chamber. The benefit of installation inside the chamber was that pressure, temperature, and humidity were stable and well-characterized. A fan circulated the air to ensure uniformity. More importantly, the range of pressures, temperatures, and water inside the chamber were very similar to what JLH experiences during aircraft measurements in UTLS field missions. The value of the AquaVIT experiment was that many of the world's leading in situ atmospheric hygrometers simultaneously measured a common source of trace water in air, and submitted data to a blind intercomparison. Close agreement was found between

instruments, in particular JLH, the Cryogenic Frostpoint Hygrometer (CFH; Voemel et al., 2007) and the AIDA facility diode laser (APicT; Ebert et al., 2005) measured water vapor within 10% over a wide range of temperatures (195 K to 243 K). This validates our approach to measuring water vapor with an open-path spectrometer, and the accuracy of the spectroscopy over a wide range of temperatures.

As part of the NGSC FTB flights, a a commercial hygrometer (FLYHT P/N: WVSS-II) accompanied JLH Mark2 on the flight of March 20, 2015. Figure 12 shows a comparison between the two instruments during an aircraft descent over Lancaster, California. Although the instruments have different time resolutions, the measurements are in general agreement to within 10%. Both JLH and FLYHT have similar detection techniques, and were originally designed under the supervision of Randy May (May, 1998), so it is not surprising that they yield similar results in the middle troposphere (700 to 250 hPa pressure). JLH has a much longer

optical path, so it has the sensitivity to measure to much lower mixing ratios in the stratosphere than FLYHT. Detailed intercomparisons with other hygrometers will be the subject of an upcoming publication.

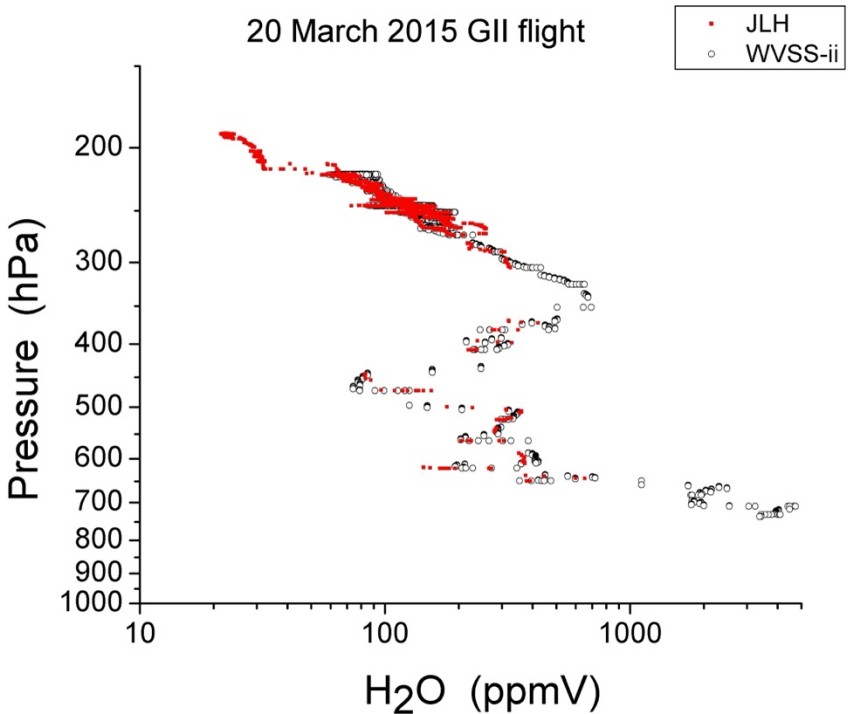

**Figure 12.** Water vapor profiles from JLH Mark2 and the FLYHT WVSS-ii sensor on March 20, 2015, during a NGSC FTB flight.

### 7. Summary

This instrument paper reports the redesigned opto-mechanical structure of the JLH Mark2 instrument, new data retrieval algorithms, and updated data analysis procedures. The mechanical redesign has significantly improved the optical stability of the instrument. We have demonstrated that misalignment and optical losses are not dominant and are no longer a limiting factor in instrument performance. JLH Mark2 has improved precision of water vapor measurements compared to earlier versions of the instrument. We present this kinematic design as an inherently stable design for atmospheric sensors that are exposed to a wide

range of temperatures.

**Data availability.** The SEAC[4]RS aircraft data used in the data analysis can be freely downloaded from the following Digital Object Identifier (DOI): SEAC4RS doi:10.5067/Aircraft/SEAC4RS/Aerosol-TraceGas-Cloud. We expect the NGSC data to be publicly released by Summer 2025. Files are available from corresponding author Robert Herman upon request:

Robert.L.Herman@jpl.nasa.gov.

**Author contributions.** Robert Herman prepared the manuscript with contributions from all coauthors and was responsible for all aspects of the JLH Mark2 as principal investigator. Robert Troy designed the mechanical structure and athermal optimal mount of JLH Mark2 and contributed the Optomechanical section of this paper, Kim Aaron designed the air foils, Isabelle Sanders and Kevin Schwarm performed the data analysis of power stability. Eric Klobas and Aaron Swanson managed the NGSC science and provided merged data sets from NGSC FTB flights, Andrew Carpenter managed the NGSC FTB and contributed its description in this paper. Scott Ozog collected and processed data from NGSC FTB flights. Keith Chin and Robert Stachnik operated the JLH Mark2 instrument in the field and downloaded data post-flight. Dejian Fu, Robert Stachnik, Keith Chin and Lance Christensen developed software components for the JLH Mark2 instrument and/or its data reduction. Ram Vasudev contributed to the spectroscopy section of this paper and Robert Jarnot contributed to the electronics section of this paper.

**Competing interests.** The authors declare that they have no conflict of interest.

**Acknowledgements**

We thank Chris Webster, Gregory Flesch, and James Margitan for helpful discussions. We thank our two reviewers, Darin Toohey and David Sayres, for extremely insightful comments that have improved this manuscript. We thank Jose Landeros and Dave Natzic for circuit assembly, cable assembly and technical support in the laboratory and field, and the aircraft crew and flight planners for making these measurements possible. We thank Kamjou Monsour for characterizing diode lasers for this project. Randy May developed the original JLH (formerly JPL $H_2O$) and W. Steve Woodward designed the original electronics used in JLH. Part of this research was performed at JPL, California Institute of Technology, formerly under a contract with NASA and more recently under a Space Act Agreement contract with Northrop Grumman Systems Corporation and NASA.

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

TABLES

**Table 1.** History of JLH aircraft flights in NASA UTLS field missions. The field mission acronyms are defined in footnotes below.

| Mission* | Timeframe | JLH Mark1 (a) | JLH Mark1 (b) 2nd Build | JLH Mark2 | Quality | Aircraft | Number of Flights | Air Base |
|---|---|---|---|---|---|---|---|---|
| NASA POLARIS | Spring-Summer 1997 | ✓ | | | Good | ER-2 | 21 | Fairbanks, Alaska |
| WAM | April, May 1998 | | ✓ | | Good | WB-57 | 3 | Houston, Texas |
| NASA ACCENT | 1999 | | ✓ | | Good | WB-57 | 8 | Houston, Texas |
| NASA SOLVE | Jan-March 2000 | ✓ | ✓ | | Good | ER-2 | 17 | Kiruna, Sweden |
| NASA CRYSTAL-FACE | 2000 | | ✓ | | Good | WB-57 | 13 | Key West, Florida |
| NASA Pre-AVE | Jan-Feb 2004 | | ✓ | | Lower S/N | WB-57 | 7 | Alajuela, Costa Rica |
| NASA MidCix | Apr-May 2004 | | ✓ | | Lower S/N | WB-57 | 7 | Houston, Texas |
| PUMA | May 2004, July 2005 | | ✓ | | Lower S/N | WB-57 | 9 | Houston, Texas |
| NASA AVE | Oct-Nov 2004, Jun-Jul 2005 | | ✓ | | Lower S/N | WB-57 | 18 | Houston, Texas |
| NASA WIIM | Jul 2005 | | ✓ | | Lower S/N | WB-57 | 4 | Houston, Texas |
| NASA CR-AVE | Jan-Feb 2006 | | ✓ | | Lower S/N | WB-57 | 16 | Alajuela, Costa Rica |
| NASA TC4-Costa Rica | Aug 2007 | | ✓ | | Very Good | WB-57 | 7 | Alajuela, Costa Rica |
| AquaVIT-1 | Oct 2007 | | ✓ | | Very Good | AIDA | 5 | Karlsruhe, Germany |
| NASA NOVICE | Sep 2008 | | ✓ | | Good | WB-57 | 3 | Houston, Texas |
| NASA MACPEX | April 2011 | | ✓ | | UT only, No LS | WB-57 | 5 | Houston, Texas |
| NASA SEAC$^4$RS | Aug, Sep 2013 | | | ✓ | Best | ER-2 | 21 | Houston, Texas |
| NGSC FTB | 2015-2023 | | | ✓ | Best | NGSC G-II | 109 | Lancaster, California |

*Definitions of acronyms follow (For the WB-57 aircraft missions see the url

http://jsc-aircraft-ops.jsc.nasa.gov/wb57/missions.html).

POLARIS: "Photochemistry of Ozone Loss in the Arctic Region In Summer", https://cloud1.arc.nasa.gov/polaris/

WAM: "WB-57 Aerosol Mission" (see NASA JSC url above);

ACCENT: "Atmospheric Chemistry of Combustion Emissions Near the Tropopause" (see NASA JSC url above);

SOLVE: "SAGE III Ozone Loss and Validation Experiment",

https://espo.nasa.gov/solve

CRYSTAL-FACE: "Cirrus Regional Study of Tropical Anvils and Cirrus Layers – Florida Area Cirrus Experiment",

https://espo.nasa.gov/crystalface

Pre-AVE: "Pre Aura Validation Experiment",

AVE: "Aura Validation Experiment" [includes AVE-Houston and maybe AVE-Houston2?]

WIIM: "Water Isotope Intercomparison Mission";

CR-AVE: "Costa Rica – Aura Validation Experiment",

https://espo.nasa.gov/ave-costarica2

TC4: "Tropical Composition, Cloud and Climate Coupling",

https://espo.nasa.gov/tc4

AquaVIT-1 (not an acronym): "International Intercomparison of atmospheric Water Vapour Instruments",

https://aquavit.icg.kfa-juelich.de/AquaVit/

NOVICE: "Newly-Operating and Validated Instruments Comparison Experiment",

https://espo.nasa.gov/novice/

MACPEX: "Mid-latitude Airborne Cirrus Properties Experiment",

https://espo.nasa.gov/macpex

SEAC$^4$RS: "Studies of Emissions and Atmospheric Composition, Clouds and Climate Coupling by Regional Surveys",

https://espo.nasa.gov/missions/seac4rs

**Table 2**. Spectroscopic parameters from HITRAN for the water absorption line at 7299.4311 cm-1 sampled in this work (Rothman et al., 2013, and Gordon et al., 2022). HITRAN 2012 parameters were utilized in data processing in 2012-2023. The latest line parameters from HITRAN 2022 are shown in the right column for comparison.

| Parameter | Name (units) | HITRAN 2012 | HITRAN 2022 |
|---|---|---|---|
| $S$ | Intensity (cm$^{-1}$/(molecule cm$^{-2}$) at 296 K) | $1.005 \times 10^{-20}$ | $1.025 \times 10^{-20}$ |
| E″ | Lower-state energy (cm$^{-1}$) | 42.3717 | 42.3717 |
| $\gamma_{air}$ | Air-broadened half width at 296 K (cm$^{-1}$ atm$^{-1}$) | 0.1032 | 0.1069 |
| $\gamma_{self}$ | Self-broadened half width at 296 K (cm$^{-1}$ atm$^{-1}$) | 0.46 | 0.48 |
| $n$ | Temperature-dependent exponent for $\gamma_{air}$ | 0.69 | 0.76 |
| δ | Air-pressure induced line shift (cm$^{-1}$ atm$^{-1}$ at 296 K) | -0.00878 | -0.0 |

**Table 3.** Summary of recent JLH laboratory calibrations

| **TS3900** **Reference** (ppmV) | **JLH** **Measurement** (ppmV) | **Drift** (ppmV) | **Fit** (ppmV) | **Offset** (ppmV) | **Offset (%)** (%) |
|---|---|---|---|---|---|
| 200.00 | 201.98 | 0.7 | 199.51 | 2.47 | 1.2% |
| 50.00 | 47.16 | 0.34 | 49.88 | -2.72 | -5.4% |
| 50.00 | 46.43 | 0.14 | 49.88 | -3.45 | -6.9% |
| 50.00 | 46.00 | 0.17 | 49.88 | -3.88 | -7.8% |
| 10.00 | 10.85 | 0.06 | 9.98 | 0.87 | 8.8% |
| 3.00 | 3.02 | 0.11 | 2.99 | 0.03 | 0.9% |