# Peer review of "JLH Mark2 - An Improved Opto-Mechanical Approach to Open-Path in situ Water Vapor Measurement in the Upper Troposphere / Lower Stratosphere"

_EGUsphere, 2024_

## Referee Comment (RC1)

Review of *"JLH Mark2 – An Improved Opto-Mechanical Approach to Open-Path in situ Water Vapor Measurement in the Upper Troposphere / Lower Stratosphere,"* by R.L. Herman, et al., Egusphere-2024-4019

By Darin Toohey, University of Colorado Boulder

This is an important contribution to the scientific literature, and AMT is an appropriate forum for publication. I recommend that the paper be published with minor revisions. The JPL group is uniquely positioned to get this information into the public domain. First, they pioneered this method for water vapor measurements from aircraft. Second, they showed the way forward that others have used. This paper documents a number of important features of the JPL aircraft instrument in various versions that are important for understanding their performance and accuracy. Although my comments are extensive, they are all straightforward for the authors to address, even the "most significant comments," and I trust them to decide how best to respond without the need for further review.

**Most significant comments**

[1] Line 217-220: There are important details in these improvements that could be explained more clearly. What, specifically, about the old components and methods resulted in more noise? Narrowing the scan presumably increased the dwell time on the main water absorption line, but this also comes at the expense of characterization of the background in the water continuum – maybe less important at very low pressures, but certainly important in the troposphere. Is it the case that changing from variable resistors to fixed resistors reduced noise, as opposed to drift in the gain? There is inherent "noise" from resistors, but I would think temperature stability is the bigger issue.

[2] Line 281-292: Although this is titled "Laser Calibration," I would suggest changing to "Laser Characterization" as it is more than calibrating. There is a lot to unpack in this short, but extremely important section that will be missed by most non-experts. It would help if the authors elaborated somewhat on why each of these matter. For example, "spectral purity" largely means how the emission of the laser is centered on a single wavelength and does not produce significant energy in "side bands." If this were to occur, there would be some residual power to the detector even when the water absorption is optically deep. There are other ways for this to occur, however. Light scattering by surfaces of the Herriott cell will reduce the effective path length even if 100% of the laser emission is in center of the water absorption line. So practically speaking, one has to estimate this contribution to scattering, which will depend on the optical depth of the absorption – i.e., it won't be the same for all concentrations of water because as the signal from the longest path is absorbed the contribution from signals from shorter paths become proportionally more important. I would suggest expanding this section a little bit to include some motivation behind each of the steps taken to characterize the laser, especially when the ultimate proof of accuracy will be a careful lab calibration that relies of some standard method for quantifying the water vapor mixing ratio over a wide range of pressures, temperatures, and water concentrations.

[3] Line 466-476: Please explain what you mean by "laser tuning rate." I assume this means the "wavelength versus time" or "versus index point." Also, why is this procedure necessary if you

are ultimately pinning your results to laboratory calibrations with known water vapor? I realize it is useful to know more about how the instrument operates, but the fact that "the response of the electronics is indeed slow, contributing to the observed somewhat large effective instrument linewidth" wouldn't seem to matter much at high pressures in the UTLS, whereas it might for a measurement at very low pressures. Maybe you can explain how RC filters, which are important for reducing high frequency electronic noise and, hence, are essential for the excellent precision you report in Figure 10, affect the accuracy of measurements at higher pressure – if at all.

[4] Line 518: What, specifically, about the "limited time response of the signal chain," causes the second negative lobe in the 2f spectrum to be "truncated?" From Figure 7 there appears to be a significant contribution from the Fabry-Perot fringe, which has a period of about 200 "index points." Or, put another way, is this "truncation" typical? Does the Fabry-Perot fringe wander with time or temperature? If this second lobe is always shifted upward, it could be the case that the phase shift in your 2f demodulation calculation is slightly off. Are you able to vary this value, or is this a non-adjustable feature of the code?

[5] Line 520-521: Do you know which "back reflection?" Is this the final reflection off the surface of the lens mounted to your detector? If so, is there a way to angle the detector or insert a wedge so that this can be minimized? Also, unless this fringe is stationary, it will contribute to your "error." Is this a significant source of uncertainty in your calculated $H_2O$ at low mixing ratios? Or is the width of the fringe much greater than the width of the $H_2O$ absorption?

[6] Line 559-561: The description of "laser drift" is a bit confusing. Are you saying that the actual temperature of the laser substrate is changing with ambient temperature for a fixed set point of the TEC control, in which case the wavelength of the laser will change? Or is it the case that with your lasers the output wavelength is determined mainly by the injection current, and the efficiency of conversion – i.e. output power versus injection current – increases with decreasing laser substrate temperature without impacting the output wavelength of the laser? Naively, I would expect the former to be the case, and you should be able to know this from the shift in the position of the absorption line.

[7] Line 577-588: The AIDA chamber demonstrations are an important "necessary" condition for the water vapor sensors that participated, meaning that one would expect the instruments to agree to within their stated uncertainties. In this case, you report "within 10% over a wide range of temperatures" which seems to be considerably larger than a back-of-envelope estimate of errors you note of 1% to 2% of various components of your careful analysis when added in quadrature. Because the JLH family of instruments are so critical to our understanding of stratospheric water, representing the only near-continuous record of in situ observations from aircraft spanning over three decades, it seems to me this paper is an excellent opportunity to "plant the flag" and make definitive statements about your best estimate for true uncertainty in the measurements without worrying about other methods which are much more difficult to "prove" from first principles. Although not required for this paper to be publishable, I think you should expand on this section, if only a little, – perhaps ½ to 1 page – to provide your best "sense" of what the true uncertainties in the water vapor mixing ratios are when the instrument is working properly. Hopefully at some point you will have a more rigorous analysis of

uncertainties, as this would be a very useful addition to the literature and historical record for JLH. Until then, would it be possible to make a summary statement about the largest sources of error in the JLH measurements and what they add up to in terms of accuracy over a typical range of pressures and temperatures for your flights?

[8] Line 590-594: This is also useful in the context of my statement about accuracy above. However, I believe the FLHYT instrument is a close relative of the JLH, having been designed and first built under the supervision of Randy May at SpectraSensors. It may be worth pointing this out, as one might expect these two instruments to yield similar results as dominant errors in them may share features that are common. Also, because JLH has a much longer optical path, it can measure to much lower mixing ratios in the stratosphere, whereas the FLHYT cuts off below 100 ppm, so it's value for UTLS measurements is somewhat limited.

**Minor Comments**

[9] Lines 16, 334, and 583: 'in situ' (no hyphen)

[10] Line 18-20: You could edit to read: "This instrument paper reports on the redesigned opto-mechanical structure of the instrument, new data retrieval algorithms, and updated data analysis procedures, along with recent laboratory and field performance and a comparison with other water vapor instruments."

[11] Line 16/22: The copy editor will know the correct style to use. You have "15 years" on Line 16 and "eight years" on Line 22.

[12] Line 29: should super saturated be one word?

[13] Line 31: "soot" may be a bit vague or too broad here, or is that your intent? For ice nucleation, I would guess metals, PAHs, others may be known better now from lab studies?

[14] Line 37: "UTLS" or "UT/LS"?

[15] Line 48: It might be helpful to state what year the mirrors were replaced.

[16] Line 64: How precisely do you know the path length, and is this invariant? In other words, is it to one part in 10,000 at all temperatures? Is it based on a ray trace or do you measure it somehow? You may want to explain in more detail – e.g.. "based on a ray trace…" and "this value is thought to be precise to one part in …" Also, accuracy matters, but I realize you calibrate the whole system with known water vapor, which effectively lumps the uncertainty in path length with the cross section of H2O.

[17] Line 84: "waterline?" Do you mean "water vapor absorption line?"

[18] Line 85-86: "28 VDC for de-ice" is a bit aircraft jargony. Maybe say "heated by a resistive element powered by 28 volts DC to reduce icing."

[19] Line 87: you will probably need to spell out ARINC somewhere – first use or in a list.

[20] Line 108: "loosely" may be too ambiguous. Maybe "approximately" is better, if you mean "we aren't sure that the focal point is exactly at midpoint."

[21] Line 117-119: Although the additional path after M0 represents only 0.8% of the total, a slowly purged volume of air in this "dead space" containing only 50 ppm could induce a +5% bias in the measurement. Is there a way to know what the H2O actually is in this dead space?

[22] Line 119-121: I realize this is getting into the weeds, but it would be useful to expand on this issue, which is critical for line-fitting at low pressures. A +/-0.01 degree shift of similar DFB lasers I have used is equivalent to xx cm-1 in wavelength, or about X5 of full-width at half-maximum. Depending on the rate of such a shift – i.e., during a scan or across multiple scans that are averaged into a single line shape profile, the gaussian shape can be distorted enough to effect "Beer-Lambert law" analysis.

[23] Line 162: It is obvious to me that "frequencies" refers to "data acquisition rates/speeds" etc., but maybe be more clear here to avoid any confusion with "frequency of laser emission."

[24] Line 165: Might be simpler to just say "The laser current scan is produced as described in May (1998)." If you intend to keep this – which would be fine – it would help to elaborate, as it may be hardware specific for the digital-to-analog converters used. For example, in our "Teensy" version of a R.D. May-like instrument described recently, we also use direct memory access, but we don't translate it using an audio codec. Rather, we can just output the stream with simple functions available in the Arduino library. So what matters isn't the "codec" per se, but rather the bit resolution – e.g., 12 bits in our case – and the update rate.

[25] Line 166: Maybe rearrange to "sawtooth-shape current ramp…" and "at a repetition rate of 8 Hz" so that it isn't confused with a wavelength specification. Also, maybe "water absorption line" instead of "feature."

[26] Line 166-167: Very minor, but maybe say "measurement method" rather than "detection method."

[27] Line 168: Perhaps "water absorbance is quantified by direct transmission (called "DC")." And why "DC" and not "DT"?

[28] Line 170: I am admittedly "old school" when it comes to digital audio. 44.1 kHz is a "CD" quality data rate. In order to generate a 128 kHz sine wave and avoid Nyquist ratio issues, I would think you would need at least 1 Mhz update rates, which isn't conventional "audio" – or is it? Maybe spell out the bit resolution and update rates specifically here, rather than saying "audio."

[29] Line 202: Maybe give the exact part numbers for the Ampro PC-104 motherboards used? "Legacy" could mean many things.

[30] Line 204: Because this is such an important snapshot of an outstanding instrument, it might be useful to provide the exact part numbers for the Vicor DC-DC converters and ripple attenuator

modules as these have changed over the years as electronics have become smaller and more energy efficient.

[31] Line 205: A lot is buried in the "custom electronics board" that I am not sure what to recommend. There is probably some proprietary information, in which case it may not be possible to expand without legal implications. But if you could, it would be useful to describe how these "custom boards" work, in principle, as this was the "magic" of the early JPL/R.D. May-designed systems. Nothing too detailed is required here, so just general approach would be useful. For example, the operating system, information about ADC resolution and data rates, communication with the main Ampro computer, etc.? Is there any storage, or is it all done in volatile memory? Or, can you refer to a paper or technical publication that describes all of this? Similar for "flash card" – what kind? How large are the files? Information that might be useful for providing some historical context, as modern storage devices have become much faster and higher capacity.

[32] Line 209: Also nit-picky – what matters more is the temperature stability – i.e., +/- 10 degrees or +/- 1 degrees – rather than the way the resistive heating elements are powered. It might be more useful to give the approximate wattage and the controller – i.e., Minco model … and the range of temperatures observed – I think you have those housekeeping values, correct?

[33] Line 209-210: This might be a place where a more precise value would be very helpful. I would guess the wavelength stability of the laser is highly dependent on this "baseplate" temperature. Are you able to maintain that to +/- 5 degrees? +/- 1 deg? Even better?

[34] Line 211-212: This is where the temperature precision really matters. I presume it is the "0.01 C" that you refer to in Line 119-121. Maybe make that clear – or, if I am wrong, clarify what the stability actually is. Again, given how important this is to wavelength stability, I assume it is something that is known and/or measured.

[35] Line 212: Is it possible to provide a rough distance between the laser and thermistor? This is important for temperature hysteresis/control.

[36] Line 215: nit picky – "signal-to-noise ratio of the water vapor measurement…"

[37] Line 222: Why is there a new board to augment the one mentioned in Line 205, rather than replace it? Wouldn't a new board be able to handle the old control while adding new capabilities? Maybe explain the idea behind the decision to not just upgrade. Or maybe this is what you intend with Lines 236-239, in which case you could bring these lines up a bit or merge these two pieces.

[38] Lines 232: Does Interagency Working Group need a reference?

[39] Line 248-249: How does filtering remove the background? Or do you mean passing the A/C component of the signal and removing the slow ramp?

[40] Line 296: This is the third instance mentioning "128 kHz" modulation. I'm not sure the repetition is necessary, so it may make sense to find the natural spot for first use of this and only mention it again if there is a need to elaborate.

[41] Line 298: Why is it "critical to know the actual modulation amplitude in spectroscopic units?" Isn't it just the case that the value for the modulation amplitude in flight be the same as the value during lab calibrations? It seems that this quantity depends on many factors, including the initial DAC driving function of the laser, detector capacitance, filtering electronics, and wavelength stability of the laser – i.e., precision of TEC temperature control. It also depends on the nature of the DFB laser substrate, which is largely empirical, and differs from laser to laser. So even if one "knows" the amplitude, there is still much to be concerned about if the value isn't stable. Maybe elaborate on why, for JLH in particular, this value is critical – i.e., is the calibration matrix keyed to a value of modulation amplitude?

[42] Line 305-307: For small optical depths it is important to document that 99.9% of the light – I presume in a non-$H_2O$-absorbing region of the spectrum – is transmitted in the final pass of the Herriott cell. However, for large optical depths the relative contribution from scattered light from earlier passes with shorter pathlengths to the detector will increase. Did you estimate what these contributions might be? E.g., how close to "null" is the signal for an optically deep line at line center? This won't matter so much for stratospheric water, but it could become a systematic error with decreasing altitude in the UT.

[43] Line 364-365: It would be useful to describe why the smaller temperature dependence of the line at 7299.4311 cm-1 is "better." Either line would still need to use a measurement of T in order to convert from H2O concentration to mixing ratio using the Ideal Gas Law, meaning that errors in T will factor proportionally into errors in mixing ratio. Unless the T dependence of the line strength is significantly greater than "proportional to T" over the range of values encountered in flight, it would seem that the uncertainties in measurements using the two different absorption lines would be comparable, especially if you are ultimately tying down your results to HiTran parameters validated with laboratory calibrations. So why is a line with a smaller sensitivity to T "better?" Is it because temperature isn't known accurately enough? Or is it simply less convenient to add another parameter into the calculation of "mixing ratio?"

[44] Lines 370-460: Sections 4.1 and 4.2 report on details that are important to document and seem fairly clear. I have no additional comments.

---

## Author Comment (AC1)

Response to reviewer RC1 of manuscript https://doi.org/10.5194/egusphere-2024-4019

JLH Mark2 - An Improved Opto-Mechanical Approach to Open-Path *in situ* Water Vapor Measurement in the Upper Troposphere / Lower Stratosphere

Robert L. Herman, Robert F. Troy, Kim M. Aaron, Isabelle Sanders, Kevin Schwarm, J. Eric Klobas, Aaron Swanson, Andrew Carpenter, Scott Ozog, Keith Chin, Lance E. Christensen, Dejian Fu, Robert F. Jarnot, Robert A. Stachnik, and Ram Vasudev

We would like to thank reviewer RC1, Prof. Darin Toohey, for his excellent and detailed comments on our manuscript. Below are our responses to each comment in order.

Major Comments

1. Line 217-220: There are important details in these improvements that could be explained more clearly. What, specifically, about the old components and methods resulted in more noise? Narrowing the scan presumably increased the dwell time on the main water absorption line, but this also comes at the expense of characterization of the background in the water continuum – maybe less important at very low pressures, but certainly important in the troposphere. Is it the case that changing from variable resistors to fixed resistors reduced noise, as opposed to drift in the gain? There is inherent "noise" from resistors, but I would think temperature stability is the bigger issue.

*Reply: When the scan was made narrower, the imod was also optimized for stratospheric conditions. Previously in 2004-2006, the laser had been overmodulated, resulting in a smaller 2f signal. Regarding the resistors, we suspect (but have not proven) that temperature changes in the variable resistors affected the gain. At line 218, we will modify the sentence to "In the signal chain gain stages, variable resistors have been replaced by precision fixed resistors to improve temperature stability of the electronic response."*

2. Line 281-292: Although this is titled "Laser Calibration," I would suggest changing to "Laser Characterization" as it is more than calibrating. There is a lot to unpack in this short, but extremely important section that will be missed by most non-experts. It would help if the authors elaborated somewhat on why each of these matter. For example, "spectral purity" largely means how the emission of the laser is centered on a single wavelength and does not produce significant energy in "side bands." If this were to occur, there would be some residual power to the detector even when the water absorption is optically deep. There are other ways for this to occur, however. Light scattering by surfaces of the Herriott cell will reduce the effective path length even if 100% of the laser emission is in center of the water absorption line. So practically

speaking, one has to estimate this contribution to scattering, which will depend on the optical depth of the absorption – i.e., it won't be the same for all concentrations of water because as the signal from the longest path is absorbed the contribution from signals from shorter paths become proportionally more important. I would suggest expanding this section a little bit to include some motivation behind each of the steps taken to characterize the laser, especially when the ultimate proof of accuracy will be a careful lab calibration that relies of some standard method for quantifying the water vapor mixing ratio over a wide range of pressures, temperatures, and water concentrations.

*Reply: We agree with the reviewer and will change the name of this section to "Laser Characterization."*

*See response to Comment 42 regarding light scattering, which we measure as negligible. To explain to non-experts why each laser parameter matters in the final result, we rewrote this paragraph:*

*"Laser characterization is necessary for accurate water vapor measurements. We first determine the absolute laser wavelength to within ~0.18 nm (1 cm$^{-1}$) with a Burleigh wavemeter. The wavelength and laser tuning rate with wavenumber are refined by using seven identified spectral lines of CH$_4$ as wavelength standards in the same spectral region as the water line of interest (Brown et al., 2013; Rothman et al., 2013). The methane line positions are known to within 10$^{-4}$ cm$^{-1}$ (Brown et al., 2013), allowing the accurate determination of the water line positions by interpolation. The laser tuning rate here refers to wavenumber (cm$^{-1}$) versus index in the laser scan, which goes from 1 to 512. For this purpose, measurements are made with JLH Mark2 in a laboratory test chamber. These weak CH$_4$ lines are measurable by filling our calibration chamber with 27.6 hPa pure CH$_4$ (with trace water). We fit a second-order polynomial to wavenumber versus index. Ultimately this wavenumber scale is needed to fit the direct absorption using fundamental physical information."*

*"Other laser parameters of interest are spectral purity, linewidth, and laser modulation amplitude. Spectral purity means the fraction of laser emission centered on a single wavelength and not in side bands (e.g., at a shifted wavelength). The measurement of water mixing ratio is directly proportional to the spectral purity. The spectral purity of the laser is verified by measuring the laser transmission through a sample at very high water vapor concentrations, which absorbs all the light at the line center wavenumber of 7299.4311 cm$^{-1}$. The laser linewidth affects the amplitude of the pp2f signal and fitting of direct absorption, so it is important to characterize. The laser linewidth is determined in the laboratory by measuring the water absorption lineshape at low pressures where the absorption linewidth is solely due to the Doppler width. If the line is wider than the*

*predicted Doppler width, it is assumed to be a convolution of the laser linewidth and Doppler width (Section 5)."*

*See response to Comment 3 below regarding laser tuning rate, moved up to line 287.*

*Lines 297-298: "The modulation amplitude is set by the user in software in electrical units (mA current to drive the laser). This amplitude is not an intrinsic laser property, but the relation between laser current and output wavelength is. It is critical for water vapor measurement accuracy to know the actual modulation amplitude in spectroscopic units (wavenumbers, cm-1) because the expected pp2f amplitude at different pressures and temperatures is sensitive to the modulation amplitude."*

3. Line 466-476: Please explain what you mean by "laser tuning rate." I assume this means the "wavelength versus time" or "versus index point." Also, why is this procedure necessary if you are ultimately pinning your results to laboratory calibrations with known water vapor? I realize it is useful to know more about how the instrument operates, but the fact that "the response of the electronics is indeed slow, contributing to the observed somewhat large effective instrument linewidth" wouldn't seem to matter much at high pressures in the UTLS, whereas it might for a measurement at very low pressures. Maybe you can explain how RC filters, which are important for reducing high frequency electronic noise and, hence, are essential for the excellent precision you report in Figure 10, affect the accuracy of measurements at higher pressure – if at all.

*Reply: Laser tuning rate refers to wavenumber (cm-1) versus index in the scan (from 1 to 512). Tuning rate is necessary to know because the wavenumber scale is needed to fit the direct absorption, which we use as checks on the pp2f. The effective instrument linewidth affects the instrument response - even at the pressures of the UTLS.*

*We will modify the sentence at line 466 (and see comment 2 above):*

*"As discussed in Section 2.3.1, laser tuning rate (wavenumber versus index point) is determined by fitting seven $CH_4$ lines surrounding the targeted water line with a second-order polynomial."*

*Laser tuning rate is defined in Section 2.3.1 with this new text in the second paragraph of Section 2.3.1:*

*"The wavelength and laser tuning rate with wavenumber are refined by using seven identified spectral lines of $CH_4$ as wavelength standards in the same spectral region as the water line of interest (Brown et al., 2013; Rothman et al., 2013). The methane line positions are known to within $10^{-4}$ cm$^{-1}$ (Brown et al., 2013), allowing the accurate determination of the water line positions by interpolation. The laser tuning rate here refers to wavenumber (cm$^{-1}$) versus index in the laser scan, which goes from 1 to 512. For this purpose,*

*measurements are made with JLH Mark2 in a laboratory test chamber. These weak $CH_4$ lines are measurable by filling our calibration chamber with 27.6 hPa pure $CH_4$ (with trace water). We fit a second-order polynomial to wavenumber versus index. Ultimately this wavenumber scale is needed to fit the direct absorption using fundamental physical information."*

4. Line 518: What, specifically, about the "limited time response of the signal chain," causes the second negative lobe in the 2f spectrum to be "truncated?" From Figure 7 there appears to be a significant contribution from the Fabry-Perot fringe, which has a period of about 200 "index points." Or, put another way, is this "truncation" typical? Does the Fabry-Perot fringe wander with time or temperature? If this second lobe is always shifted upward, it could be the case that the phase shift in your 2f demodulation calculation is slightly off. Are you able to vary this value, or is this a non-adjustable feature of the code?

*Reply: We have revised our explanation for the asymmetric negative lobes, which are also seen by other TDLAS. This effect is due to a shift between amplitude response and frequency response. In line 518 in the original manuscript, we speculated that the time response of the signal chain is the cause of asymmetric negative lobes (Figure 7). We will change the sentence at line 518 to:*

*"It has commonly been observed in the tunable diode laser (TDL) community that 2f spectra have asymmetric negative lobes, and other researchers have found reasons other than electronic filtering that can cause this artifact (e.g., Goldenstein et al., 2014)."*

*New reference:*

*Goldenstein, C. S., Strand, C. L., Schultz, I. A., Sun, K., Jeffries, J. B., and Hanson, R. K., Fitting of calibration-free scanned-wavelength-modulation spectroscopy spectra for determination of gas properties and absorption lineshapes, Applied Optics, 53(3), 2014, http://dx.doi.org/10.1364/AO.53.000356.*

*This "truncation" of the second negative lobe is typical, and the Fabry-Perot interference fringe is stationary. The fringe does not wander with either time or temperature (see below response to Comment 5).*

5. Line 520-521: Do you know which "back reflection?" Is this the final reflection off the surface of the lens mounted to your detector? If so, is there a way to angle the detector or insert a wedge so that this can be minimized? Also, unless this fringe is stationary, it will contribute to your "error." Is this a significant source of uncertainty in your calculated H2O at low mixing ratios? Or is the width of the fringe much greater than the width of the H2O absorption?

*Reply: We do not know which "back reflection" causes this fringe, but it corresponds to a free spectral range of a few centimeters. Since we do not know the source, we do not want to speculate which reflection in the manuscript. The reviewer is correct that angling the detector or inserting a wedge could minimize this back reflection in future work. The width of this stationary fringe is much greater than the width of the water absorption line so it does not contribute significantly to the error. Therefore, we will modify the text in line 521, "Fig. 7 shows curved 2f baseline due to a Fabry-Perot interference fringe from a short back-reflection in the optical system. The fringe width is greater than the width of the water absorption line. Since JLH response is tied to the laboratory measurements, this fringe does not contribute a significant source of uncertainty to calculated water vapor at low mixing ratios."*

6. Line 559-561: The description of "laser drift" is a bit confusing. Are you saying that the actual temperature of the laser substrate is changing with ambient temperature for a fixed set point of the TEC control, in which case the wavelength of the laser will change? Or is it the case that with your lasers the output wavelength is determined mainly by the injection current, and the efficiency of conversion – i.e. output power versus injection current – increases with decreasing laser substrate temperature without impacting the output wavelength of the laser? Naively, I would expect the former to be the case, and you should be able to know this from the shift in the position of the absorption line.

*Reply: The laser temperature is changing with ambient temperature for a fixed set point of the TEC control. We will remove the last sentence of this paragraph at line 560-561, and insert new text, "The laser wavelength is drifting during flight because the temperature of the laser substrate is changing with ambient temperature - even for a fixed set point of the TEC control. Line locking would solve the issues but was not implemented in this compact system."*

7. Line 577-588: The AIDA chamber demonstrations are an important "necessary" condition for the water vapor sensors that participated, meaning that one would expect the instruments to agree to within their stated uncertainties. In this case, you report "within 10% over a wide range of temperatures" which seems to be considerably larger than a back-of-envelope estimate of errors you note of 1% to 2% of various components of your careful analysis when added in quadrature. Because the JLH family of instruments are so critical to our understanding of stratospheric water, representing the only near-continuous record of in situ observations from aircraft spanning over three decades, it seems to me this paper is an excellent opportunity to "plant the flag" and make definitive statements about your best estimate for true uncertainty in the measurements without worrying about other methods which are much more difficult to "prove" from first principles. Although not required for this paper to be publishable, I think you should expand on this section, if only a little, – perhaps ½ to 1 page – to provide your best "sense" of what the true uncertainties in the water vapor mixing ratios are when the instrument is working properly. Hopefully at some point you will have a more rigorous analysis of

uncertainties, as this would be a very useful addition to the literature and historical record for JLH. Until then, would it be possible to make a summary statement about the largest sources of error in the JLH measurements and what they add up to in terms of accuracy over a typical range of pressures and temperatures for your flights?

*Reply:*

*In Section 2.3.2 (see below), we will provide a summary statement about the largest sources of error in the JLH measurements and what they add up to in terms of accuracy over a typical range of pressures and temperatures for recent flights. We will also revise Section 2.3.2 (at line 326) to show recent calibration results:*

*"Below, we describe a recent example of JLH laboratory calibration. Fig. 4a and 4b show time series of JLH measurements within the lab calibration chamber as 200 ppmv and 50 ppmv, respectively, are flowed through the chamber.*

[Figure]

***Figure 4.*** *Time series of JLH water vapor measurement inside the laboratory calibration chamber with continuous flow of air/water vapor mixture from the TS3900 humidity generator. (a) Left plot is 200 ppmv input at 57.2 hPa total pressure and 295.4 K temperature, (b) Right plot is 50 ppmv input at 120.3 hPa total pressure and 296.4 K temperature.*

*Fig. 4c demonstrates the linearity of JLH response versus reference gas standard from the TS3900 from 3 ppmv to 200 ppmv. This is also shown in Table 3 along with the difference in percent and ppmv. Generally, results are within 9% but there is a pressure dependence as shown in Fig. 4d. This pressure dependence has been documented in our previous laboratory calibrations. At higher and lower pressures, the response changes systematically.*

[Figure]

[Figure]

*Figure 4.* *(c) Left plot demonstrates that the relation of JLH water measurement to TS3900 water source is close to linear. A 1:1 line (dashed line) is also shown. (d) Right plot is JLH water measurement inside lab calibration chamber at different total pressures while the TS3900 provides a constant source of 50 ppmv water vapor at room temperature.*

**Table 3. Summary of recent JLH laboratory calibrations**

| TS3900 Reference | JLH Measurement | Drift | Fit | Offset | Offset (%) |
|---|---|---|---|---|---|
| (ppmv) | (ppmv) | (ppmv) | (ppmv) | (ppmv) | (%) |
| 200.00 | 201.98 | 0.7 | 199.51 | 2.47 | 1.2% |
| 50.00 | 47.16 | 0.34 | 49.88 | -2.72 | -5.4% |
| 50.00 | 46.43 | 0.14 | 49.88 | -3.45 | -6.9% |
| 50.00 | 46.00 | 0.17 | 49.88 | -3.88 | -7.8% |
| 10.00 | 10.85 | 0.06 | 9.98 | 0.87 | 8.8% |
| 3.00 | 3.02 | 0.11 | 2.99 | 0.03 | 0.9% |

*The best estimate of JLH total error for NASA SEAC[4]RS and more recent FTB flights is 12%, adding in quadrature the following terms. The largest error source is 9% in water measurement due to mismatch of the actual lineshape and the predicted lineshape as manifested as pressure-dependence (Fig. 4d). We estimate 5% uncertainty in water*

*measurement from uncertainty in the temperature dependence of the actual lineshape and 6% from scaling to laboratory calibrations (Fig. 4c and Table 3). There is 1% error in water measurement due to uncertainty in the in-flight temperature and pressure measurements. Finally, only 0.1% uncertainty in water measurement is due to uncertainty in the pathlength, and a fraction of a percent from other error sources such as wavelength, laser spectral purity, light scattering, et cetera.*

8. Line 590-594: This is also useful in the context of my statement about accuracy above. However, I believe the FLHYT instrument is a close relative of the JLH, having been designed and first built under the supervision of Randy May at SpectraSensors. It may be worth pointing this out, as one might expect these two instruments to yield similar results as dominant errors in them may share features that are common. Also, because JLH has a much longer optical path, it can measure to much lower mixing ratios in the stratosphere, whereas the FLHYT cuts off below 100 ppm, so it's value for UTLS measurements is somewhat limited.

*Reply: The reviewer is correct that both the FLHYT and JLH instruments were designed and first built under the supervision of Randy May. We will add two sentences to line 592, "Both JLH and FLHYT have similar detection techniques, and were originally designed under the supervision of Randy May (May, 1998), so it is not surprising that they yield similar results in the middle troposphere (700 to 250 hPa pressure). JLH has a much longer optical path, so it has the sensitivity to measure to much lower mixing ratios in the stratosphere than FLHYT."*

Minor Comments

9. Lines 16, 334, and 583: 'in situ' (no hyphen)

*Reply: We will consistently use "in situ" with no hyphen.*

10. Line 18-20: You could edit to read: "This instrument paper reports on the redesigned opto-mechanical structure of the instrument, new data retrieval algorithms, and updated data analysis procedures, along with recent laboratory and field performance and a comparison with other water vapor instruments."

*Reply: We will change lines 18-20 to: "This instrument paper reports on the redesigned opto-mechanical structure of the instrument, new data retrieval algorithms, and updated data analysis procedures, along with recent laboratory and field performance and a comparison with other water vapor instrument."*

11. Line 16/22: The copy editor will know the correct style to use. You have "15 years" on Line 16 and "eight years" on Line 22.

*Reply: We will write out the number in line 16: "fifteen years".*

12. Line 29: should super saturated be one word?

*Reply: Yes, we will change this to one word, "supersaturated."*

13. Line 31: "soot" may be a bit vague or too broad here, or is that your intent? For ice nucleation, I would guess metals, PAHs, others may be known better now from lab studies?

*Reply: We misspoke, soot is unnecessary for contrail formation. According to D. Cziczo (pers. comm.), "The current thinking is that contrails mainly form homogeneously, meaning the RH gets high enough to basically activate any ambient particles. These might be the soot in the engine exhaust but also any ambient aerosol that gets entrained [Testa et al., 2024]." We will modify this sentence at line 31 to "In such an environment, aircraft engine combustion elevates the relative humidity high enough for homogeneous ice nucleation in persistent contrails and aircraft induced cirrus clouds (e.g., Testa et al., 2024, and references therein)." and add a new reference:*

*Testa, B., Durdina, L., Edebeli, J., Spirig, C., and Kanji, Z. A.: Simulated contrail-processed aviation soot aerosols are poor ice-nucleating particles at cirrus temperatures, Atmos. Chem. Phys., 24, 10409–10424, https://doi.org/10.5194/acp-24-10409-2024, 2024.*

14. Line 37: "UTLS" or "UT/LS"?

*Reply: We will change this to "UTLS" to be consistent with widespread usage.*

15. Line 48: It might be helpful to state what year the mirrors were replaced.

*Reply: We will modify the sentence in lines 48-49 to, "For example, extensive exposure to rocket plumes deteriorated the mirror coatings so that it became necessary to replace the mirrors in June 2007 prior to the NASA TC$^4$ mission (Table 1)."*

16. Line 64: How precisely do you know the path length, and is this invariant? In other words, is it to one part in 10,000 at all temperatures? Is it based on a ray trace or do you measure it somehow? You may want to explain in more detail – e.g.. "based on a ray trace…" and "this value is thought to be precise to one part in …" Also, accuracy matters, but I realize you calibrate the whole system with known water vapor, which effectively lumps the uncertainty in path length with the cross section of H2O.

*Reply: The path length is calculated from ray trace for the well-defined optical path of fifty passes between mirrors of known radius of curvature (219.66 mm). The path length is known to one part in 10,000 over a 100 K temperature range because the length of the long*

*axis of the multi-pass cell is defined by invar spacing rods, with a coefficient of thermal expansion of 10^-6 K^-1. The sentence in Lines 64-65 points to Section 2.1, so we will add a sentence at line 119: "The total absorption path length is calculated based on a ray trace, knowing the Herriott configuration for fifty passes between mirrors of known radii of curvature (Altmann et al., 1981; Herriott et al., 1964). This value is thought to be precise to one part in 10,000 over the ambient temperature range of 100 K in flight because the length of the long axis of the multi-pass cell is defined by invar spacing rods, which have a coefficient of thermal expansion of 10^-6 K^-1."*

17. Line 84: "waterline?" Do you mean "water vapor absorption line?"

*Reply: We have adjusted our wording to indicate that we are referring to a vertical coordinate on the air vehicle. For example, waterline 0 refers to the horizontal plane at the datum and waterline -2 refers to the horizontal plane two inches below the datum. In this case, the TAT sensor was mounted at the same waterline index as the stock TAT probe, but on the alternate side of the air vehicle. New line 84: "… compared to the production, i.e. same fuselage station and vertical waterline coordinate, but on the port side of the fuselage rather than starboard."*

18. Line 85-86: "28 VDC for de-ice" is a bit aircraft jargony. Maybe say "heated by a resistive element powered by 28 volts DC to reduce icing."

*Reply: We have revised the text in lines 85 to 86 to " The selected TAT probe is a dual mandrel design, three inches in length to measure outside the boundary layer, and resistively heated to prevent ice buildup."*

19. Line 87: you will probably need to spell out ARINC somewhere – first use or in a list.

*Reply: ARINC-429 was developed by Aeronautical Radio, Incorporated in 1977 and remains the most common avionics digital information transfer system used in commercial aviation. Although the name ARINC is derived from the name Aeronautical Radio, Incorporated, the data specification is published as ARINC-429.*

*We have revised the text in line 87:*

*"The airframe static pressure ports connect to Gulfstream Air Data Computers that drive an ARINC-429 (Aeronautical Radio, Incorporated) digital information transfer system (Martinec et al., 2015) which is then displayed to the pilots."*

*We also add a citation to the ARINC-429 specification.*

*Martinec, D. A., Buckwalter, S. P., and  Zurawski, R.:ARINC Specification 429 Mark 33 Digital Information Transfer System. In Industrial Communication Technology Handbook (2nd ed., pp. 45-1-45–16). CRC Press. https://doi.org/10.1201/b17365-47, 2015.*

20. Line 108: "loosely" may be too ambiguous. Maybe "approximately" is better, if you mean "we aren't sure that the focal point is exactly at midpoint."

*Reply: We will change the wording to "approximately"*

21. Line 117-119: Although the additional path after M0 represents only 0.8% of the total, a slowly purged volume of air in this "dead space" containing only 50 ppm could induce a +5% bias in the measurement. Is there a way to know what the H2O actually is in this dead space?

*Reply: The additional optical path is flushed, so water vapor remains relatively close to environmental mixing ratios. This optical path is an open cavity between the hole in mirror $M_0$ and the free stream between the air foil and laserhead housing (Figure 1) and air flows through this as explained below. We will add several sentences at line 117:*

*"Air flow in this additional path prevents any trapped air that could bias the water vapor measurements. The mirror airfoils were designed to maintain the ambient pressure external to the aircraft between the mirrors. In between the mirror airfoil and the laser housing, the geometry of the parts forms a venturi. At typical flight air speeds, the pressure there will be lower than the ambient; there will be air flow from the hole in mirror $M_0$ through the airfoils due to this differential pressure."*

22. Line 119-121: I realize this is getting into the weeds, but it would be useful to expand on this issue, which is critical for line-fitting at low pressures. A +/-0.01 degree shift of similar DFB lasers I have used is equivalent to xx cm-1 in wavelength, or about X5 of full-width at half-maximum. Depending on the rate of such a shift – i.e., during a scan or across multiple scans that are averaged into a single line shape profile, the gaussian shape can be distorted enough to effect "Beer-Lambert law" analysis.

*Reply: at line 119-121, we will add a pointer to Section 2.2.1 "(see Section 2.2.1)." Laser drift is discussed in Section 2.2.1 (see Comments 34 and 35).*

23. Line 162: It is obvious to me that "frequencies" refers to "data acquisition rates/speeds" etc., but maybe be more clear here to avoid any confusion with "frequency of laser emission."

*Reply: We removed the word "frequencies" from line 162 text for clarity. The new text is: "Minor changes made to data acquisition and instrument implementation are described below."*

24. Line 165: Might be simpler to just say "The laser current scan is produced as described in May (1998)." If you intend to keep this – which would be fine – it would help to elaborate, as it may be hardware specific for the digital-to-analog converters used. For example, in our "Teensy" version of a R.D. May-like instrument described recently, we also use direct memory access, but we don't translate it using an audio codec. Rather, we can just output the stream

with simple functions available in the Arduino library. So what matters isn't the "codec" per se, but rather the bit resolution – e.g., 12 bits in our case – and the update rate.

*Reply: We will change line 165 to "The laser current scan is produced as described in May (1998) using an audio codec integrated circuit in continuous direct memory access. However, in our implementation, we do not utilize the codec for translation and thus are not limited by the codec data rate but rather by the bit resolution - 16 bits - and an update rate of 256 kHz (equivalent to the 2f demodulation frequency)."*

25. Line 166: Maybe rearrange to "sawtooth-shape current ramp…" and "at a repetition rate of 8 Hz" so that it isn't confused with a wavelength specification. Also, maybe "water absorption line" instead of "feature."

*Reply: We agree with the suggested change. We will modify line 166 to:*

*"The TDL is driven with a sawtooth-shape current ramp at a ramp repetition rate of 8Hz to scan across the targeted water absorption transition. Data are acquired at 4 kHz (512 points per ramp)."*

26. Line 166-167: Very minor, but maybe say "measurement method" rather than "detection method."

*Reply: We will make the suggested change to the text.*

27. Line 168: Perhaps "water absorbance is quantified by direct transmission (called "DC")." And why "DC" and not "DT"?

*Reply: We mean direct transmission (DT) here. "DC" refers to the direct current channel of the signal chain (as opposed to AC) (May, 1998). We will modify the text at line 168 to "water absorbance is quantified by direct transmission".*

28. Line 170: I am admittedly "old school" when it comes to digital audio. 44.1 kHz is a "CD" quality data rate. In order to generate a 128 kHz sine wave and avoid Nyquist ratio issues, I would think you would need at least 1 Mhz update rates, which isn't conventional "audio" – or is it? Maybe spell out the bit resolution and update rates specifically here, rather than saying "audio."

*Reply: Coupled to comment 24. The response there addresses both comments.*

29. Line 202: Maybe give the exact part numbers for the Ampro PC-104 motherboards used? "Legacy" could mean many things.

*Reply: We will add text that "This is the legacy Ampro PC-104 motherboard from May (1998), which is an AMPRO 486D-02506 (S/N 6472175B), A60739, manufactured in 1996 and running the DOS operating system. It formerly flew on the JLH short-path instrument on the NASA DC-8 platform in CAMEX-3 (2001)."*

30. Line 204: Because this is such an important snapshot of an outstanding instrument, it might be useful to provide the exact part numbers for the Vicor DC-DC converters and ripple attenuator modules as these have changed over the years as electronics have become smaller and more energy efficient.

*Reply: We will add text that " Aircraft 28 V DC power is converted to +5 V and +12 V DC using Vicor VI-JW0-IY and VI-JW1-IY DC-DC converters with VI-AWW-IU input attenuation module and VI-RAM-I1 ripple attenuation module (from the mid-1990s) with additional filtering and overvoltage protection."*

31. Line 205: A lot is buried in the "custom electronics board" that I am not sure what to recommend. There is probably some proprietary information, in which case it may not be possible to expand without legal implications. But if you could, it would be useful to describe how these "custom boards" work, in principle, as this was the "magic" of the early JPL/R.D. May-designed systems. Nothing too detailed is required here, so just general approach would be useful. For example, the operating system, information about ADC resolution and data rates, communication with the main Ampro computer, etc.? Is there any storage, or is it all done in volatile memory? Or, can you refer to a paper or technical publication that describes all of this? Similar for "flash card" – what kind? How large are the files? Information that might be useful for providing some historical context, as modern storage devices have become much faster and higher capacity.

*Reply: We cannot expand on proprietary information about the electronics boards. We will add text:*

*"Largely this PC-104 stack has preserved the electronics from the mid-1990s technology (May, 1998) for consistency in measurements. Laser drive and data acquisition are carried out using the MicroNIR Primary Board Rev. 5 (MCD 97-0200C), from 1998. This board uses the Crystal Semiconductor CS4215 chip, which led to a more compact, low-power implementation than would have been provided by any other approach available in the mid-1990s. Temperature and engineering data are acquired by the Secondary Board PF-194V-0 (MCD 50706-B). Files are written to a flash card, legacy M-Systems DiskOnChip 2000 DIP, 128 MB, P/N MD2202-D128-X (extended temperature range), Ver. 5.1.2."*

32. Line 209: Also nit-picky – what matters more is the temperature stability – i.e., +/- 10 degrees or +/- 1 degrees – rather than the way the resistive heating elements are powered. It might be more useful to give the approximate wattage and the controller – i.e., Minco model … and the range of temperatures observed – I think you have those housekeeping values, correct?

*Reply: We will modify the sentence about the electronics heater in lines 208-209: "The electronics are temperature-stabilized for greater accuracy and stability. The aluminum baseplate of the aluminum electronics box is stabilized to 298.0 K by a Minco strip heater (HK5493R6.1L24E) in pulse-width modulation control by the Secondary Board (May, 1998).*

*The observed temperature is 298.0 +/- 0.1 K except for transient fluctuations of up to 0.5 K for 2 minutes when the aircraft rapidly changes altitude."*

33. Line 209-210: This might be a place where a more precise value would be very helpful. I would guess the wavelength stability of the laser is highly dependent on this "baseplate" temperature. Are you able to maintain that to +/- 5 degrees? +/- 1 deg? Even better?

*Reply: We will modify the sentence about the aluminum heat-sink in lines 209-210: "The aluminum heat-sink on one side of the thermoelectric cooler (TEC) is heated to 288.0 +/- 0.1 K by a Minco strip heater (HK5163R44.0L24B) controlled by a commercial temperature controller (MPT2500, Wavelength Electronics Corp.). The measured stability of temperature is +/-0.1 K during flight when the laser housing is colder than 288 K."*

34. Line 211-212: This is where the temperature precision really matters. I presume it is the "0.01 C" that you refer to in Line 119-121. Maybe make that clear – or, if I am wrong, clarify what the stability actually is. Again, given how important this is to wavelength stability, I assume it is something that is known and/or measured.

*Reply: We will modify the sentence about the laser/detector pair in lines 211-212: "For greater stability, the laser and detector on the aluminum cold block on the other side of the TEC are further temperature stabilized by active heating/cooling of the TEC. The TEC temperature is regulated by a subminiature temperature controller (HY5610, Hytek). This controller operates by proportional/integral control to stabilize the aluminum cold block to +/-0.01 K."*

35. Line 212: Is it possible to provide a rough distance between the laser and thermistor? This is important for temperature hysteresis/control.

*Reply: Referring back to the original drawing, the distance between laser and thermistor through the aluminum cold block is approximately 21 mm. We will add a sentence to line 212, "The HY5610 senses the resistance (corresponding to temperature) of a thermistor mounted within the aluminum cold block. This thermistor is 21 mm away from the laser housing in the cold block. This distance may affect temperature hysteresis and control (see Section 6.1)."*

36. Line 215: nit picky – "signal-to-noise ratio of the water vapor measurement…"

*Reply: We will edit the text accordingly to "signal-to-noise ratio of the water vapor measurement…"*

37. Line 222: Why is there a new board to augment the one mentioned in Line 205, rather than replace it? Wouldn't a new board be able to handle the old control while adding new capabilities? Maybe explain the idea behind the decision to not just upgrade. Or maybe this is what you intend with Lines 236-239, in which case you could bring these lines up a bit or merge these two pieces.

*Reply: At line 222, we will add, "Through trial and error, we found that adding capabilities to the CPU board interfered with the C code controlling the instrument. As a result, we added the separate interface processor board (IPB) dedicated to "on-the-fly" processing and broadcasting the data packets."*

38. Lines 232: Does Interagency Working Group need a reference?

*Reply: Yes, in line 232 we will cite Webster et al. (2024) for Interagency Working Group:*

*Webster, C., Dominguez, R., Freudinger, L., Hill, J., Oolman, L., Meyers, J., Sorenson, C., Tomlinson, J., Sullivan, D.:IWG1 ASCII Packet Definition. http://n2t.net/ark:/85065/d70k2dtg, 2024.*

39. Line 248-249: How does filtering remove the background? Or do you mean passing the A/C component of the signal and removing the slow ramp?

*Reply: What we mean by "filtering removing the background" is that filtering passes the AC component of the signal and removes DC offsets caused by the slow ramp. Lines 248-249 will be rewritten,*

*"The time-varying signal is detected at frequency 2f = 256 kHz and filtered to remove DC offsets caused by the slow laser current ramp."*

40. Line 296: This is the third instance mentioning "128 kHz" modulation. I'm not sure the repetition is necessary, so it may make sense to find the natural spot for first use of this and only mention it again if there is a need to elaborate.

*Reply: We agree line 296 is out of place. In line 170, we will keep the first mention of f = 128 kHz modulation. We repeat f = 128 kHz in line 248 for clarity (see above). We will remove the sentence in line 296, "For very sensitive detection with second harmonic detection in wavelength modulation spectroscopy (May and Webster, 1993; May, 1998), the laser wavelength is also modulated in addition to being tuned by superimposing a small sinusoidal modulation at a frequency f = 128 kHz on the tuning ramp."*

41. Line 298: Why is it "critical to know the actual modulation amplitude in spectroscopic units?" Isn't it just the case that the value for the modulation amplitude in flight be the same as the value during lab calibrations? It seems that this quantity depends on many factors, including the initial DAC driving function of the laser, detector capacitance, filtering electronics, and wavelength stability of the laser – i.e., precision of TEC temperature control. It also depends on the nature of the DFB laser substrate, which is largely empirical, and differs from laser to laser. So even if one "knows" the amplitude, there is still much to be concerned about if the value isn't stable. Maybe elaborate on why, for JLH in particular, this value is critical – i.e., is the calibration matrix keyed to a value of modulation amplitude?

*Reply: We will modify the sentence at lines 298-299 to, "It is critical for water vapor measurement accuracy to know the actual modulation amplitude in spectroscopic units because the expected pp2f amplitude at different pressures and temperatures in the look-up tables is sensitive to the modulation amplitude."*

42. Line 305-307: For small optical depths it is important to document that 99.9% of the light – I presume in a non-H2O-absorbing region of the spectrum – is transmitted in the final pass of the Herriott cell. However, for large optical depths the relative contribution from scattered light from earlier passes with shorter pathlengths to the detector will increase. Did you estimate what these contributions might be? E.g., how close to "null" is the signal for an optically deep line at line center? This won't matter so much for stratospheric water, but it could become a systematic error with decreasing altitude in the UT.

*Reply: We measured negligible signal from scattered light at line center. We will add a sentence to line 307, "At large optical depths (e.g., measurements on the ground in humid air), no contribution from scattered light was detected."*

43. Line 364-365: It would be useful to describe why the smaller temperature dependence of the line at 7299.4311 cm-1 is "better." Either line would still need to use a measurement of T in order to convert from H2O concentration to mixing ratio using the Ideal Gas Law, meaning that errors in T will factor proportionally into errors in mixing ratio. Unless the T dependence of the line strength is significantly greater than "proportional to T" over the range of values encountered in flight, it would seem that the uncertainties in measurements using the two different absorption lines would be comparable, especially if you are ultimately tying down your results to HiTran parameters validated with laboratory calibrations. So why is a line with a smaller sensitivity to T "better?" Is it because temperature isn't known accurately enough? Or is it simply less convenient to add another parameter into the calculation of "mixing ratio?"

*Reply: We thank the reviewer for bringing this to our attention. Lines 364-365 are incorrect and will be removed because the T-dependence of line strength is accurately calculated from HITRAN parameters for the entire range of expected pressures and temperatures. Furthermore, 7299 cm-1 line has a slightly larger T-dependence of line strength (relative to 7294 cm-1) due to higher lower-state energy E'' (cm-1). We will remove the entire paragraph lines 362-367 because it is out of place here in Section 3.*

44. Lines 370-460: Sections 4.1 and 4.2 report on details that are important to document and seem fairly clear. I have no additional comments.

*Reply: No response required here. Referee approves of Sections 4.1 and 4.2.*

---

## Author Comment (AC2)

Response to reviewer RC2 of manuscript https://doi.org/10.5194/egusphere-2024-4019

JLH Mark2 - An Improved Opto-Mechanical Approach to Open-Path *in situ* Water Vapor Measurement in the Upper Troposphere / Lower Stratosphere

Robert L. Herman, Robert F. Troy, Kim M. Aaron, Isabelle Sanders, Kevin Schwarm, J. Eric Klobas, Aaron Swanson, Andrew Carpenter, Scott Ozog, Keith Chin, Lance E. Christensen, Dejian Fu, Robert F. Jarnot, Robert A. Stachnik, and Ram Vasudev

We would like to thank reviewer RC2, Dr. David Sayres, for his excellent and detailed comments on our manuscript. Below are our responses to each comment in order.

Minor Comments:

1. Line 28: addition should additional

*Reply: We will change this word to "additional".*

2. Line 37: Be consistent with UTLS or UT/LS

*Reply: We will change this (and all uses of this word) to "UTLS" to be consistent with widespread usage.*

3. Lines 83-90: Where are these temperature and pressure measurements compared to JLH. Could you comment on the suitability of these measurements for the water absorption. Do you expect there to be T and P differences between where these measurements are made and the Herriott cell?

*Reply: These instruments are situated several meters distant from each other in both water line coordinate and fuselage station coordinate; however, Computational Fluid Dynamics (CFD) modeling indicates that these instruments sample beyond the boundary layer in the freestream.*

*At approximate air speeds of 250 m/s, instantaneous variations in atmospheric water vapor mixing ratio, static air temperature, and static air pressure at the integrated instrumentation locations are assumed to be trivial in comparison to the variations in these*

*parameters in the total atmospheric volume sampled during the data integration period of 1 Hz. Consequentially, the atmospheric volumetric average parameter value is expected to be identical beyond instrumental precision at each location.*

*We have revised the text to clarify that these instruments sample the freestream.*

*[line 79] "and flights of opportunity (2015-2023). For these flights, JLH Mark2 was mounted on top of the aircraft fuselage where it samples the atmospheric freestream (Figure 1b)."*

*[line 86] "The probe was routed to an instrumentation system for data collection. Static pressure was also sampled outside the boundary layer. The ..."*

4. Line 107: Can you comment on boundary layer depth of the fuselage.

*Reply: CFD alpha and beta sweeps encompassing expected parameters for cruise, ascent, and descent at relevant Mach numbers and altitudes were performed. In no condition investigated did the boundary layer extend to the optical measurement.*

*We note that this conclusion is also provided on line 330 of the original manuscript. We revise line 107 to provide the reader with an earlier indication that the instrument optics lay in the free-stream. The new sentence on line 107 is:*

*"The optical open-path multipass absorption cell of JLH Mark2 is mounted external to the aircraft and situated beyond the expected boundary layer in order to avoid contamination or..."*

5. Line 117: This additional pathlength seems potentially an important source of error. Do you believe this air to be flushed in the same way the air between the mirrors? Have the same temperature and pressure? I realize it's only 0.8% of the pathlength, but if this volume is "dead air" it will have much higher mixing ratio of water.

*Reply: The additional optical path is flushed, so water vapor remains relatively close to environmental mixing ratios. This optical path is an open cavity between the hole in mirror $M_0$ and the free stream between the air foil and laserhead housing (Figure 1) and air flows through this as explained below. We will add several sentences at line 117:*

"The mirror airfoils were designed to maintain the air density between the two mirrors equal to the free stream density (see Section 4.2). Outside the mirror airfoil, between the $M_0$ mirror airfoil and the laser housing, the geometry of the parts forms a venturi. At typical flight air speeds, the pressure between the airfoil and the laser housing will be lower than the ambient; there will be air flow from the hole in mirror $M_0$ through the airfoils due to this differential pressure."

6. Line 119: How is the pathlength calculated? Is this by ray trace or knowing the angle that the light bouncing between the mirrors?

*Reply: The pathlength is calculated by ray trace. We will add an additional sentence to line 119:*

*"The total absorption path length is calculated based on a ray trace, knowing the Herriott configuration for fifty passes between mirrors of known radii of curvature (Altmann et al., 1981; Herriott et al., 1964). This value is thought to be precise to one part in 10,000 over the ambient temperature range of 100 K in flight because the length of the long axis of the multi-pass cell is defined by invar spacing rods, which have a coefficient of thermal expansion of $10^{-6}$ $K^{-1}$."*

7. Line 166: 8 Hz is a little ambiguous here. You mean that you have 8 ramps a second, but the actual rate of data acquisition is much higher. Probably best to have another sentence that specifies the acquisition rate and ramp rate and remove the 8 Hz at the end of the sentence.

*Reply: To make this more clear, we will modify line 166 to:*

*"The TDL is driven with a sawtooth-shape current ramp at a ramp repetition rate of 8 Hz to scan across the targeted water absorption transition. Data are acquired at 4 kHz (512 points per ramp)."*

8. Line 168: What is DC mean here, direct current, or did you mean DT?

*Reply: We mean direct transmission (DT) here. "DC" refers to the direct current channel of the signal chain (as opposed to AC) (May, 1998). We will modify the text at line 168 to "water absorbance is quantified by direct transmission".*

9. Line 211: Be good to mention how well this works. What was the flight stability of the laser temperature thermistor.

*Reply: The temperature stability of the laser temperature thermistor (and the aluminum housing) is +/-0.01 deg C as already mentioned in line 120.*

10. Line 249: should this be 'background noise'?

*Reply: What we mean by "background" is that filtering passes the AC component of the signal and removes DC offsets caused by the slow ramp. Lines 248-249 will be rewritten,*

*"The time-varying signal is detected at frequency 2f = 256 kHz and filtered to remove DC offsets caused by the slow laser current ramp."*

11. Line 287: I'm assuming the interpolation is linear, but inherently a laser's tuning rate with current does not have to be linear, so it might be worth noting whether using the methane lines you find it linear over this scan range.

*Reply: The interpolation is a second-order polynomial that is very close to linear. We will clarify in the text that we fit the wavelength scale versus index to a second-order polynomial using seven identified methane lines and the known water line (pure methane with trace water in our calibration chamber). First, we rewrote this entire paragraph to better explain:*

*"Laser characterization is necessary for accurate water vapor measurements. We first determine the absolute laser wavelength to within ~0.18 nm (1 $cm^{-1}$) with a Burleigh wavemeter. The wavelength and laser tuning rate with wavenumber are refined by using seven identified spectral lines of $CH_4$ as wavelength standards in the same spectral region as the water line of interest (Brown et al., 2013; Rothman et al., 2013). The methane line positions are known to within $10^{-4}$ $cm^{-1}$ (Brown et al., 2013), allowing the accurate determination of the water line positions by interpolation. The laser tuning rate here refers to wavenumber ($cm^{-1}$) versus index in the laser scan, which goes from 1 to 512. For this purpose, measurements are made with JLH Mark2 in a laboratory test chamber. These weak $CH_4$ lines are measurable by filling our calibration chamber with 27.6 hPa pure $CH_4$ (with trace water). We fit a second-order polynomial to wavenumber versus index. Ultimately this wavenumber scale is needed to fit the direct absorption using fundamental physical information."*

*"Other laser parameters of interest are spectral purity, linewidth, and laser modulation amplitude. Spectral purity means the fraction of laser emission centered on a single wavelength and not in side bands (e.g., at a shifted wavelength). The measurement of water mixing ratio is directly proportional to the spectral purity. The spectral purity of the laser is verified by measuring the laser transmission through a sample at very high water vapor concentrations, which absorbs all the light at the line center wavenumber of 7299.4311 $cm^{-1}$. The laser linewidth affects the amplitude of the pp2f signal and fitting of direct absorption, so it is important to characterize. The laser linewidth is determined in the laboratory by measuring the water absorption lineshape at low pressures where the absorption linewidth is solely due to the Doppler width. If the line is wider than the predicted Doppler width, it is assumed to be a convolution of the laser linewidth and Doppler width (Section 5)."*

*Second, we will modify the sentence at line 466:*

*"As discussed in Section 2.3.1, laser tuning rate (wavenumber versus index point) is determined by fitting seven CH$_4$ lines surrounding the targeted water line with a second-order polynomial."*

12. Line 292: It would seem that this last sentence flows more naturally right after the discussion of determining the water line position (ending in the middle of line 287).

*Reply: We agree with the reviewer and will move the text up to line 287 (see comment 11 above).*

13. Lines 309 – 312: I find this procedure confusing or at least its description. Is the HITRAN database parameters being used to fit the direct absorption spectra? When you say you use the ratio, do you mean for the field data you scale the direct absorption measurements by this scale factor? So ultimately the standard that you are using to calibrate your instrument is the Thunder Scientific.

*Reply: Yes, the standard that I am using to calibrate JLH is the Thunder Scientific. All the JLH measurements are scaled to the Thunder Scientific (TS3900), so we have revised the calibration text at lines 309-312:*

"The instrument response is characterized through continuous flow of an accurately known air/water vapor mixture through the JLH sample cell when it is inside the laboratory calibration chamber (see below). JLH water vapor measurements are calculated as in the field data. The ratio of known source water data to JLH water data is used as the instrument response factor and is used to scale the field measurements."

"By 2007, improved calibration methods were developed, which allowed for laboratory measurements of sample atmospheres in test chambers with water volume mixing ratios as low as 2 parts-per-million by volume (ppmV) (Troy, 2007). The source of the air-water mixture for determining the instrument response mentioned above is a commercial Thunder Scientific 3900 (TS3900) low-humidity generator (https://www.thunderscientific.com/), which can provide air with stable water mixing ratios from less than 1 ppmV to 12,000 ppmV. The carrier gas is ultrazero air from cylinders. The TS3900 uses a technique that generates air with a constant water mixing ratio to an accuracy of 0.1 K frostpoint (corresponding to better than 1% accuracy of mixing ratio). The technique is a primary standard recognized by NIST and used by commercial hygrometer manufacturers to check their instruments. It depends solely on accurate measurements of pressure and temperature to achieve a constant mixing ratio of water vapor in air flowing over a saturator. By scaling the JLH data to the TS3900, the standard we are using to calibrate the instrument is the TS3900."

"For an open-path instrument such as JLH, it is necessary to place the laser/detector housing and sample cell inside a test chamber. The JLH test chamber used for laboratory calibration is a 304 Corrosion Resistant Steel (CRES) box vacuum chamber manufactured by the Kurt J. Lesker Company, and coated on the inside with a hydrophobic coating of Fluoropel® 1302IBA, manufactured by Cytonix Inc. of Beltsville, Maryland (Troy, 2007). For calibration of the JLH instrument in the lab, an air/water vapor mixture generated by the TS3900 flows continuously through the CRES chamber at constant pressure and temperature. The sampled air mixture is checked both upstream and downstream of JLH with a reference Vaisala DM500X precision surface-acoustic wave (SAW) hygrometer. Typically, half of the flow is directed through JLH and the rest through the Vaisala hygrometer, which has a quoted accuracy of 0.3 K frostpoint (corresponding to 2% accuracy of water mixing ratio)."

14. Line 312: I'd be careful of saying you are canceling the uncertainties in the HITRAN database. There may be many sources of error for fitting the direct absorption line. Line purity for one, which you mention. The choice of which line shape parameters you are using. At higher concentrations or for your studies using pure water self broadening can be important and also choice of lineshape (Voigt versus Galatry for example). How well you know your tuning rate and how linear it is. For the purposes of measuring water vapor in the atmosphere using methane lines to derive a tuning rate and position is perfectly adequate, but may miss some small nonlinearities in the laser.

*Reply: We agree with the author that there are many sources of error for fitting the direct absorption line. We will remove the sentence at Line 312 ("This effectively cancels out the errors due to uncertainties in the line strength in the HITRAN database.").  The tuning rate is known well because it is fit by a polynomial using seven methane lines. Uncertainties are addressed in our response to your final comment (below). Our methodology uses a Voigt lineshape for curve-fitting, but ultimately the measurements over the range of concentrations (3 to 1000 ppmv) are scaled to match the Thunder Scientific.*

*Furthermore, we will remove the paragraph at lines 527-534 because it overstates the accuracy of fitting the direct absorption line.*

15. Line 361: Not sure why this section is called Laser stability. 'Absorption line selection' perhaps.

*Reply: We agree that the topic of Section 3.2 is "absorption line selection" and not "laser stability." This is a short paragraph that belongs better in section 2.3.1 Laser Characterization, so we will move the text to the start of Section 2.3.1 (line 282).*

16. Line 362: Given that everywhere else you write 'water', I'd change H2O to water. Unless there is some more subtle point you are trying to make that I've missed.

*Reply: We agree and will change "H2O" to "water"*

17. Line 466: laser tuning rate here refers to cm-1 versus time or versus current? It should be defined somewhere. You also mention it earlier, so perhaps defining it there.

*Reply: Laser tuning rate refers to wavenumber (cm-1) versus index in the scan (from 1 to 512). This is also addressed in our response to comment 11 above, where we rewrote Section 2.3.1.*

*Second, we will modify the sentence at line 466:*

*"As discussed in Section 2.3.1, laser tuning rate (wavenumber versus index point) is determined by fitting seven $CH_4$ lines surrounding the targeted water line with a second-order polynomial."*

18. Lines 466 – 476: The discussion of line widths is a little confusing. On line 472 you say "the Gaussian is a convolution of Doppler-broadened linewidth and laser linewidth, then we calculate an effective instrument linewidth ..." Given the first clause of that sentence, what you are calculating is the laser linewidth. You then go on to say that the instrument line width is now a convolution of the true laser linewidth and electronic broadening which would seem to invalidate the presumption you make on line 472. I think it would be simpler to say that the difference between the Gaussian fit and the calculated Doppler broadening is dominated by the laser linewidth and any electronic broadening. You believe (I don't think you say that you've measured the laser linewidth independently) the laser linewidth is smaller and therefore think that the electronics are broadening the line. Usually if the electronics are the issue they act as an RC filter of the data. That would not broaden the line symmetrically around the line center. You can also characterize the electronic time constant by chopping the laser light or cutting the current quickly and looking at the exponential decay on an oscilloscope.

*Reply: The reviewer is correct that electronics would act as an RC filter of the data to produce an asymmetric line shape. We do not observe an asymmetric direct transmission line shape, but rather a symmetric Gaussian line shape (e.g., Figure 6). To produce a Gaussian line shape, we hypothesize that random fluctuations in either laser temperature or laser current are responsible. We have not proven this, however, so we will rewrite this paragraph similar to what this reviewer has recommended:*

*Lines 466-476:*

*"As discussed in Section 2.3.1, laser tuning rate (wavenumber versus index point) is determined by fitting seven $CH_4$ lines surrounding the targeted water line with a second-order polynomial. The instrument line shape is then characterized by comparison of a low-pressure absorption feature with the expected profile of a Doppler line shape. Figure 6 shows an experimental line shape obtained with the Herriott cell in a laboratory test chamber at a temperature of 297.54 K with pure water vapor at a pressure of $9 \times 10^{-2}$ hPa. The observed absorptance, $(I-I_o)/I$, is only ~0.134, so the sample is optically thin and the line shape is due to only Doppler broadening. This Gaussian is 20% broader than the calculated Doppler-broadened lineshape for this measured temperature. We hypothesize that the difference between the Gaussian fit and the calculated Doppler broadening is dominated by fluctuations of the laser wavelength (associated with either temperature or dithering noise), but have not fully diagnosed the issue."*

19. Line 518: Could you explain what you mean by the limited time response of the signal chain?

*Reply: We have revised our explanation for the asymmetric negative lobes, which are also seen by other TDLAS. This effect is due to a shift between amplitude response and frequency response. In line 518 in the original manuscript, we speculated that the time response of the signal chain is the cause of asymmetric negative lobes (Figure 7). We will change the sentence at line 518 to:*

*"It has commonly been observed in the tunable diode laser (TDL) community that 2f spectra have asymmetric negative lobes, and other researchers have found reasons other than electronic filtering that can cause this artifact (e.g., Goldenstein et al., 2014)."*

*New reference:*

*Goldenstein, C. S., Strand, C. L., Schultz, I. A., Sun, K., Jeffries, J. B., and Hanson, R. K., Fitting of calibration-free scanned-wavelength-modulation spectroscopy spectra for determination of gas properties and absorption lineshapes, Applied Optics, 53(3), 2014, http://dx.doi.org/10.1364/AO.53.000356.*

20. Figure 10: The y axis label says 'and Pressure'. I think that is a typo.

*Reply: Good catch. We will change the y axis label to "Water (ppmv)"*

21. Section 6. Performance: In all this discussion, you never state your accuracy. You talk about the calibration procedure in in section 2.3.2, but I expected at some point a statement or graph showing water vapor in your lab sample flow as determined by the Thunder Scientific versus water vapor mixing ratio calculated by JLH as you would do in flight. A plot of the points showing the linearity over a couple order of magnitude of water. A

difference plot and statement of accuracy in percent or ppmv. I think this is critical for an instrument paper. Comparison with other instruments is not sufficient.

*Reply: here is the addition of accuracy discussion to the revised paper:*

*New text at line 323:*

*"For an open-path instrument such as JLH, it is necessary to place the laser/detector housing and sample cell inside a test chamber. The JLH test chamber used for laboratory calibration is a 304 Corrosion Resistant Steel (CRES) box vacuum chamber manufactured by the Kurt J. Lesker Company, and coated on the inside with a hydrophobic coating of Fluoropel® 1302IBA, manufactured by Cytonix Inc. of Beltsville, Maryland (Troy, 2007). For calibration of the JLH instrument in the lab, an air/water vapor mixture generated by the TS3900 flows continuously through the CRES chamber at constant pressure and temperature."*

*We will revise also Section 2.3.2 (at line 326) to discuss accuracy and to show recent calibration results:*

*"Below, we describe a recent example of JLH laboratory calibration. Fig. 4a and 4b show time series of JLH measurements within the lab calibration chamber as 200 ppmv and 50 ppmv, respectively, are flowed through the chamber.*

[Figure]

***Figure 4.*** *Time series of JLH water vapor measurement inside the laboratory calibration chamber with continuous flow of air/water vapor mixture from the TS3900 humidity generator. (a) Left plot is 200 ppmv input at 57.2 hPa total pressure and 295.4 K temperature, (b) Right plot is 50 ppmv input at 120.3 hPa total pressure and 296.4 K temperature.*

*Fig. 4c demonstrates the linearity of JLH response versus reference gas standard from the TS3900 from 3 ppmv to 200 ppmv. This is also shown in Table 3 along with the difference in percent and ppmv. Generally, results are within 9% but there is a pressure dependence as shown in Fig. 4d. This pressure dependence has been documented in our previous laboratory calibrations. At higher and lower pressures, the response changes systematically.*

[Figure]

[Figure]

**Figure 4.** *(c) Left plot demonstrates that the relation of JLH water measurement to TS3900 water source is close to linear. A 1:1 line (dashed line) is also shown. (d) Right plot is JLH water measurement inside lab calibration chamber at different total pressures while the TS3900 provides a constant source of 50 ppmv water vapor at room temperature.*

**Table 3. Summary of recent JLH laboratory calibrations**

| TS3900 Reference | JLH Measurement | Drift | Fit | Offset | Offset (%) |
|---|---|---|---|---|---|
| (ppmv) | (ppmv) | (ppmv) | (ppmv) | (ppmv) | (%) |
| 200.00 | 201.98 | 0.7 | 199.51 | 2.47 | 1.2% |
| 50.00 | 47.16 | 0.34 | 49.88 | -2.72 | -5.4% |
| 50.00 | 46.43 | 0.14 | 49.88 | -3.45 | -6.9% |
| 50.00 | 46.00 | 0.17 | 49.88 | -3.88 | -7.8% |
| 10.00 | 10.85 | 0.06 | 9.98 | 0.87 | 8.8% |
| 3.00 | 3.02 | 0.11 | 2.99 | 0.03 | 0.9% |

*The best estimate of JLH total error for NASA SEAC⁴RS and more recent FTB flights is 12%, adding in quadrature the following terms. The largest error source is 9% in water measurement due to mismatch of the actual lineshape and the predicted lineshape as manifested as pressure-dependence (Fig. 4d). We estimate 5% uncertainty in water*

*measurement from uncertainty in the temperature dependence of the actual lineshape and 6% from scaling to laboratory calibrations (Fig. 4c and Table 3). There is 1% error in water measurement due to uncertainty in the in-flight temperature and pressure measurements. Finally, only 0.1% uncertainty in water measurement is due to uncertainty in the pathlength, and a fraction of a percent from other error sources such as wavelength, laser spectral purity, light scattering, et cetera.*